# Sensitivity of outcome instruments in a priori selected patient groups after traumatic brain injury: Results from the CENTER-TBI study

Nicole von Steinbuechel[1], Katrin Rauen[2,3], Amra Covic[1], Ugne Krenz[1], Fabian Bockhop[1], Isabelle Mueller[1,4], Katrin Cunitz[1], Suzanne Polinder[5], Ewout W. Steyerberg[5,6], Johannes Vester[7], Marina Zeldovich[1]*, the CENTER-TBI participants investigators[¶]

1 Institute of Medical Psychology and Medical Sociology, University Medical Center Göttingen, Göttingen, Germany, 2 Department of Geriatric Psychiatry, Psychiatric Hospital Zurich, University of Zurich, Zurich, Switzerland, 3 Institute for Stroke and Dementia Research, University Hospital, LMU Munich, Munich, Germany, 4 Columbia University Medical Center, Department of Psychiatry, New York, NY, United States of America, 5 Department of Public Health, Erasmus MC, University Medical Center Rotterdam, Rotterdam, The Netherlands, 6 Department of Biomedical Data Sciences, Leiden University Medical Center, Leiden, The Netherlands, 7 Department of Biometry and Clinical Research, idv Data Analysis and Study Planning, Gauting, Germany

¶ The complete list of the CENTER-TBI participants and investigators and can be found in the Acknowledgments.
* marina.zeldovich@med.uni-goettingen.de

**Data Availability Statement:** All relevant data are available upon request from CENTER-TBI, and the authors are not legally allowed to share it publicly.

## Abstract

Traumatic brain injury (TBI) can negatively impact patients' lives on many dimensions. Multiple instruments are available for evaluating TBI outcomes, but it is still unclear which instruments are the most sensitive for that purpose. This study examines the sensitivity of nine outcome instruments in terms of their ability to discriminate within and between specific patient groups, selected a priori as identified from the literature, at three different time points within a year after TBI (i.e., 3, 6, and 12 months post injury). The sensitivity of the instruments to sociodemographic (sex, age, education), premorbid (psychological health status), and injury-related (clinical care pathways, TBI and extracranial injury severity) factors was assessed by means of cross-sectional multivariate Wei-Lachin analyses. The Glasgow Outcome Scale Extended (GOSE)–the standard in the field of TBI for measuring functional recovery–demonstrated the highest sensitivity in most group comparisons. However, as single functional scale, it may not be able to reflect the multidimensional nature of the outcome. Therefore, the GOSE was used as a reference for further sensitivity analyses on more specific outcome scales, addressing further potential deficits following TBI. The physical component summary score (PCS) of the generic health-related quality of life (HRQOL) instruments (SF-36v2/-12v2) and the TBI-specific HRQOL instruments (QOLIBRI/-OS) were most sensitive in distinguishing recovery after TBI across all time points and patient groups, followed by the RPQ assessing post-concussion symptoms and the PHQ-9 measuring depression. The SF-36v2/-12v2 mental component summary score and the GAD-7 measuring anxiety were less sensitive in several group comparisons. The assessment of the functional recovery status combined with generic HRQOL (the PCS of the SF-12v2), disease-specific HRQOL (QOLIBRI-OS), and post-concussion symptoms (RPQ) can provide a

The authors confirm that they received no special access privileges to the data. CENTER-TBI is committed to data sharing and in particular to responsible further use of the data. Hereto, we have a data sharing statement in place: https://www.center-tbi.eu/data/sharing. The CENTER-TBI Management Committee, in collaboration with the General Assembly, established the Data Sharing policy, and Publication and Authorship Guidelines to assure correct and appropriate use of the data as the dataset is hugely complex and requires help of experts from the Data Curation Team or Bio-Statistical Team for correct use. This means that we encourage researchers to contact the CENTER-TBI team for any research plans and the Data Curation Team for any help in appropriate use of the data, including sharing of scripts. Requests for data access can be submitted online: https://www.center-tbi.eu/data. The complete Manual for data access is also available online: https://www.center-tbi.eu/files/SOP-Manual-DAPR-2402020.pdf.

**Funding:** This study was funded by the European Union 7th Framework programme [grant no. 602150]. Additional funding was obtained from the Hannelore Kohl Stiftung (Germany), OneMind (USA), Integra LifeSciences Corporation (USA), and NeuroTrauma Sciences (USA). The funders of the study had no role in study design, data collection, data analysis, data interpretation, or writing of the report.

**Competing interests:** The authors have read the journal's policy and have the following competing interests: Johannes Vester is a senior biometric consultant of idv Datenanalyse und Versuchsplanung and received no personal fees related to the submitted work. Integra LifeSciences Corporation (USA) and NeuroTrauma Sciences (USA) provided additional financial support in respect to data curation. This does not alter our adherence to PLOS ONE policies on sharing data and materials. There are no patents, products in development or marketed products associated with this research to declare.

sensitive, comprehensive, yet time-efficient evaluation of the health status of individuals after TBI in different patient groups.

## Introduction

Traumatic brain injury (TBI) is a relevant personal health and economic burden worldwide, which is characterized by multi-level medical [1], neuropsychiatric [2], cognitive [3], emotional [4], and psychosocial sequelae [5]. If the consequences of TBI are left untreated, they may not only have long-term adverse effects on the health status of those affected, but also on their health-related quality of life (HRQOL) [6] and the quality of life of their family members [7, 8].

Multiple instruments are available for evaluating outcomes after TBI. However, it is still unclear which instruments are the most sensitive for this purpose. Using sensitive instruments is crucial if valid conclusions are to be drawn about the clinical relevance of outcomes (e.g., the presence or extent of impairment), allowing relevant treatments and therapies to be selected. Sensitivity is defined as the ability of an instrument to detect changes and/or differences, for example, in the health status of different patient groups, thus characterizing the clinical usefulness of an outcome measure. A distinction is made between cross-sectional sensitivity (i.e., the discriminative ability at a given point in time) and longitudinal sensitivity (i.e., the sensitivity to change, which is also called responsiveness) [9]. There are several ways of assessing sensitivity (e.g., using effect size, relative efficiency, the receiver operating characteristic [ROC] curve, or measurement sensitivity [10]). Sensitivity can also be compared with a currently preferred outcome measure, a gold standard assessing the outcome of a particular disease [11]. Combining such a standard with other outcome measures–particularly measures based on self-reports capturing patients' subjective views of a disease–complements the evaluation of problems and symptoms that may be overlooked when only the clinicians' perspective is adopted.

Until now, the extended version of the Glasgow Outcome Scale (GOSE) [12] has been the only core instrument listed among the Common Data Elements (CDE) recommendations on outcome measures in the field of adult TBI [13, 14]. The GOSE is a clinician-reported outcome (ClinRO) measure of global functioning and recovery for rating the aggregated effects of central and peripheral injuries on disability and global functional recovery [12]. Despite its widespread use, several studies have found evidence of item redundancy [15], information deficiency [15], item inefficiency (being insensitive to minimal yet relevant functional changes regarding activities of daily living after TBI), producing ceiling effects [16], as well as the fact that it may not capture the full extent of problems that patients suffer from after TBI [17, 18]. Further evidence for the sensitivity of the GOSE could therefore strengthen its use as a clinical standard in the field of TBI.

In recent years, a growing body of TBI research into patient-reported outcome measures (PROMs) has shown that cognitive disturbances [19, 20], post-concussion symptoms [4], depressive and anxiety disorders [21, 22], posttraumatic stress disorder symptoms [23, 24] and deficits in HRQOL [6] may also limit wellbeing and global recovery after TBI [25–27]. Thus, supplementing the assessment of the recovery status as measured by the GOSE with information on other outcome domains potentially affected by the sequelae of TBI could provide a more comprehensive picture of the patient's health status.

A wide range of literature, including systematic reviews, meta-analyses, cross-sectional and longitudinal multinational studies, has addressed the question of identifying protective and

risk factors for TBI outcome. Previous research has shown that outcomes may be influenced by the sociodemographic and clinical characteristics of individuals affected by TBI. There are controversial results concerning men or women having better outcomes, possibly in association with the premorbid health status or injury severity [28, 29]. In addition, individuals aged 65 years and older are at a higher risk of mortality and unfavorable outcomes after TBI compared to younger individuals [30]. Individuals with a lower pre-injury level of education tend to have worse cognitive outcomes after TBI and lower probability of a satisfactory return to work and life [31]. Furthermore, the premorbid health status [32–34] and injury-related factors (e.g., different mechanisms of brain trauma [35–37], severity of brain injury [38, 39], or presence of extracranial injuries or major trauma [40]) may influence the outcome after TBI. A comparison of outcomes of uncomplicated and complicated mild TBI patients based on the combination of the Glasgow Coma Scale (GCS) [41] and findings from computed tomography (CT) scans [42] has shown that individuals after a complicated mild TBI had worse functional outcomes, decreased HRQOL, and a higher symptom burden compared with those who had experienced an uncomplicated mild TBI [39, 43]. Overall, lower functional recovery, reduced generic and TBI-specific HRQOL, and higher symptom burden (i.e., anxiety, depression, post-traumatic stress disorder, and post-concussion symptoms) were repeatedly associated with female gender [21, 29], higher age [44–46], lower education [25, 47, 48], the presence of premorbid psychological problems [4, 45, 49–51], being discharged home from the emergency room [52] or being admitted to the ICU [43, 53, 54], as well as having more severe extracranial injuries or polytrauma [38, 43, 53, 55–57], and higher TBI severity [24, 38, 56, 58, 59] (see S1 Table for a more detailed overview). Hence, analyzing the sensitivity of outcome instruments to patient groups based on these characteristics can assist in selecting the appropriate instruments. This may contribute to better clinical decision-making and personalized treatment.

Given the impact of TBI on different domains of health and life, and considering the heterogeneity of potential risk and protective factors, a sensitive multidimensional approach is needed to identify the short- and long-term effects of the injury. To date, only the sensitivity of individual instruments used in the field of TBI has been assessed, if at all. Systematic analyses of the multivariate sensitivity of the instruments measuring outcome domains concerning patient groups selected a priori in the field of TBI is still scarce.

To fill this gap, the sensitivity of the PROMs that assess these domains needs to be investigated with reference to several relevant patient groups, which are known from the TBI literature, and with reference to functional recovery. We therefore aim to investigate the multidimensional cross-sectional sensitivity of selected outcome instruments using a patient-centered, group-based diagnostic approach. This approach includes the analysis of sensitivity at three different time points (i.e., 3, 6, and 12 months) as the sensitivity of the instruments can differ depending on the time of assessment post TBI. The aims of our study are:

1. To analyze the sensitivity of nine outcome instruments measuring different dimensions of health to six patient groups selected a priori based on sociodemographic, premorbid, and injury-related characteristics:

   a. ClinRO: *Functional recovery* after TBI (GOSE combined with information from assessments using its questionnaire version GOSE-Q [60])

   b. PROMs: *Generic HRQOL* (Short-Form 36 and 12 – Version 2; SF-36v2 [61]; SF-12v2 [62]); *TBI-specific HRQOL* (Quality of Life after Traumatic Brain Injury and its short form, the overall scale; QOLIBRI [63, 64], QOLIBRI-OS [65]), *anxiety* (Generalized Anxiety Disorder-7; GAD-7 [66]), *depression* (Patient Health Questionnaire-9; PHQ-9 [67]), *posttraumatic stress disorder* (Posttraumatic Stress Disorder Checklist for DSM-5;

PCL-5 [68]), and *post-concussion symptoms* (Rivermead Post-Concussion Symptoms Questionnaire; RPQ [69])

2. To analyze the sensitivity of the PROMs with respect to the standard in the field of TBI measuring functional recovery–the GOSE–in six patient groups selected a priori;

3. To provide general recommendations for clinicians and researchers on selection of the most sensitive instruments concerning a priori patient group criteria for outcome evaluation during a year after TBI, as well as recommendations for three specific time points.

## Materials and methods

### Participants

From December 9, 2014 until December 17, 2017 study participants were recruited at 63 centers across 18 European countries and in Israel for the prospective, multicenter, longitudinal, observational cohort study Collaborative European NeuroTrauma Effectiveness Research (CENTER-TBI; EC grant 602150; clinicaltrials.gov NCT0221022). The inclusion criteria for study participation were a clinical diagnosis of TBI, written informed consent (obtained from participants or from their legal representatives), presentation within 24 hours after injury, and an indication for computed tomography (CT) scanning. Individuals were assigned to three strata corresponding to their primary clinical pathways: all patients were admitted to the emergency room (ER), then either discharged, or admitted to a hospital ward (ADM), or to the intensive care unit (ICU). Data were collected either at the hospital, through face-to-face or telephone interviews, or via postal mail. Further study details can be found elsewhere [52].

The core study sample consisted of 4,509 individuals [52]. In this study, we focused on participants aged 16 years and above who had completed at least one outcome measure at the three-, six-, and twelve-months post-TBI assessments. Data were retrieved from the Core 2.0 data set using the data access tool Neurobot.

### Ethical approval

The CENTER-TBI study was conducted in accordance with all relevant laws of the EU where directly applicable or having a direct effect, and all relevant laws of the countries in which the recruiting sites were located, including but not limited to, the relevant privacy and data protection laws and regulations (the "Privacy Law"), the relevant laws and regulations on the use of human materials, and all relevant guidance relating to clinical studies from time to time in force including, but not limited to, the ICH Harmonized Tripartite Guideline for Good Clinical Practice (CPMP/ICH/135/95) ("ICH GCP") and the World Medical Association Declaration of Helsinki entitled "Ethical Principles for Medical Research Involving Human Subjects". Written informed consent was obtained for all patients recruited to the core data set of CENTER-TBI and documented in the e-CRF. Ethical approval was obtained for each recruiting site. The list of sites, ethical committees, approval numbers and approval dates can be found on the project's website https://www.center-tbi.eu/project/ethical-approval.

### Sociodemographic, premorbid, and injury-related data

The sensitivity of the outcome instruments was examined using a priori selected groups covering sociodemographic, premorbid, and injury-related characteristics derived from previous studies. S1 Table provides an overview of these characteristics influencing outcome domains (i.e., functional recovery, generic and disease-specific HRQOL, anxiety, depression, PTSD, and post-concussion symptoms) after TBI. The selected factors were found to be both

significant and clinically relevant in several studies concerning a single outcome domain after TBI. For this reason, considering them when selecting instruments may have substantial benefits concerning diagnosis and treatment planning. Our multivariate analyses were therefore stratified according to the following sociodemographic characteristics: sex (male/female), age (<65/$\geq$ 65 years), and education (primary and less/at least secondary). Premorbid health status and injury-related characteristics were assessed using the following information collected at the time of study enrollment: individuals' psychological health status before the injury (emotional disorders, treatment for any mental health problems, or hospital admission for psychiatric reasons; absent/present), clinical pathways (ER/ward/ICU), and total injury severity score (ISS; with the cut-off values <10 indicating mild injury vs. $\geq$10 including moderate, severe and profound injuries) [70] as measured by the Abbreviated Injury Scale (AIS) [71]. TBI severity was determined based on the GCS together with the information on CT findings, resulting in the following groups: uncomplicated mild (GCS $\geq$ 13 and no CT abnormalities), complicated mild (GCS $\geq$ 13 and visible CT abnormalities), moderate ($9 \leq$ GCS $\leq 12$), and severe (GCS $\leq 8$) TBI.

## Instruments

The selection of the instruments used in the present study was informed by the CDE recommendations on TBI outcome measures [13, 14]. Instruments lacking translations in the languages of the countries participating in the CENTER-TBI study were translated, and linguistically and psychometrically validated [72, 73].

**Functional recovery status after TBI.** Functional recovery after TBI was rated using the *Glasgow Outcome Scale Extended (GOSE-Interview)* [12] and its self- or proxy-rated version, the *Glasgow Outcome Scale Extended–Questionnaire version (GOSE-Q)* [60]. The GOSE is a 19-question clinician-rated interview evaluating functional status and recovery of individuals after TBI. The GOSE-Q covers similar aspects to the GOSE and includes 14 items with a different response format that can be answered either by the affected individual or by their proxy. A rating scale was established for both versions of the instrument. The GOSE covers eight levels of recovery (1 = dead, 2 = vegetative state, 3/4 = lower/upper severe disability, 5/6 = lower/upper moderate disability, 7/8 = lower/upper good recovery) and the GOSE-Q seven levels, as no differentiation is possible between 2 = vegetative state and 3 = lower severe disability.

To avoid loss of information, missing GOSE values (14%–21% depending on the time of assessment) were centrally imputed using the ratings derived either from the GOSE-Q or interviewer ratings. The imputing procedure is described elsewhere [74]. Since the GOSE-Q cannot distinguish between vegetative state and lower severe disability, these categories were combined into one. This combined information on the recovery status of the participants is therefore referred to as GOSE/-Q.

**Patient-reported outcome measures (PROMs).** The *Generalized Anxiety Disorder-7 (GAD-7)* [66] questionnaire assesses seven symptoms of a generalized anxiety disorder using a four-point Likert scale (from "not at all" to "nearly every day") with a recall period of two weeks. The total score ranges from 0 to 21, with values of 10 and above indicating clinically relevant anxiety [66].

The *Patient Health Questionnaire-9 (PHQ-9)* [67] captures the severity of major depression using nine items based on DSM-IV ([75]) criteria on a four-point Likert scale (from "not at all" to "nearly every day") with a recall period of two weeks. The total score ranges from 0 to 27, with a score of 10 and above indicating clinically relevant depression [67, 76].

The *Posttraumatic Stress Disorder Checklist for the DSM (PCL-5)* [68] assesses 20 symptoms characterizing PTSD based on criteria of the fifth edition of the Diagnostic and Statistical

Manual of Mental Disorders (DSM-5) [77] with a recall period of a week or a month. The items are rated on a five-point Likert scale (from "not at all" to "extremely"). The total score ranges from 0 to 80, with a cut-off score of 33 indicating clinically relevant impairment [51].

The *Rivermead Post-Concussion Symptoms Questionnaire (RPQ)* [69] evaluates 16 emotional, cognitive, and somatic post-concussion symptoms. Individuals report how much they suffered from each of the symptoms over the past 24 hours compared with their condition before TBI, using a five-point Likert scale (from "not experienced at all" to "a severe problem"). The total score ranges from 0 to 64, with higher values indicating greater impairment. For clinical screening, a cut-off score of 12 can be applied [78].

TBI-specific and generic HRQOL were assessed using the long and short forms of the respective instruments:

The *Quality of Life after Brain Injury Scale (QOLIBRI)* [63, 64] is a TBI-specific instrument comprising 37 items and using a five-point Likert response scale (from "not at all" to "very"). The items cover the following six domains: cognition, self, daily life and autonomy, social relationships, emotions, and physical problems. The total score is transformed into a percentage ranging from 0 to 100, with higher values being associated with better HRQOL [64]. In general, a score less than 60 indicates impaired HRQOL [79]. Country-specific reference values can provide more specific information [80].

The *Quality of Life after Brain Injury–Overall Scale (QOLIBRI-OS)* [65]. In the short version of the QOLIBRI with six items, physical conditions, cognition, emotions, daily life and autonomy, social relationships, and current and future prospects are assessed on a five-point Likert scale (from "not at all" to "very"). In general, a score below 52 indicates impaired HRQOL [79]. The use of country-specific reference values is recommended where available [81].

The *36-item Short Form Health Survey–Version 2 (SF-36v2)* [61] measures the generic subjective health status using 36 items with various response formats (dichotomous "yes/no" to polytomous five-point Likert scale responses) on eight scales. Scores range from 0 to 100, with higher values associated with better HRQOL. These can be transposed into T-values using normative data. A value below 47 indicates impairment (based on data for the U.S. general population) [61]. Items can be summarized to form the physical component summary score (*PCS*) and the mental component summary score (*MCS*). To determine impaired generic HRQOL in this multicenter study, a cut-off of 40 (i.e., 50-1*SD*) was applied.

The *12-Item Short Form Survey–Version 2 (SF-12v2)* [62] uses twelve items derived from the SF-36v2 which can also be summed up into the *PCS* and *MCS*. Both scores range from 0 to 100, with higher values associated with better HRQOL. Scores can be transposed into T-values using normative data. The authors recommend using country- and group-specific cut-off values [62, 82]. To avoid loss of information, missing values in the SF-12v2 items were centrally replaced by the values derived from the respective items of the SF-36v2 and combined with reported data. To determine impaired generic HRQOL in this multicenter study, a cut-off of 40 (i.e., 50–1*SD*) was applied.

**Statistical analyses.** Descriptive analyses of the sociodemographic, premorbid, and injury-related characteristics of the participants were reported. To account for the nature of the GOSE ratings, all statistical approaches chosen were appropriate for ordinal data. Spearman correlations investigated the strength of associations between the outcome domains. Effect sizes were classified as being small (0.10), medium (0.30), and large (0.50) [83, 84]. Medium to high associations between the outcome instruments warrant conducting multivariate analyses.

A non-parametric Wei-Lachin [85] approach was applied to examine the sensitivity of the outcome instruments. This approach allows multiple outcome comparisons to be performed simultaneously relative to a control group, which is suitable for continuous and ordinal data

[86]. For each instrument, the sensitivity in distinguishing between and within patient groups was assessed using the Mann-Whitney (MW) effect size, which is equivalent to the area under the receiver-operating characteristic (ROC) curve [87]. The MW effect size varies from 0 to 1, with 0.50 indicating group equality. It represents the probability that a randomly chosen participant from the first patient group of interest (e.g., male after an uncomplicated mild TBI) has a better outcome (e.g., TBI-specific HRQOL assessed using the QOLIBRI) compared with the second group of interest (e.g., female after an uncomplicated mild TBI). The strength of the sensitivity was evaluated using conventional cut-off values indicating small ($0.36 \leq MW \leq 0.64$), medium (beyond 0.36 or 0.64, but greater than 0.29 or less than 0.71), and large (less than or equal to 0.29 or greater than or equal to 0.71) effects [84, 88]. Based on these, a MW = 0.29, corresponding to a large effect size, indicates that males have better outcomes than females with respect to TBI-specific HRQOL after an uncomplicated mild TBI. Large effect size represents a high ability of the QOLIBRI to discriminate between males and females after uncomplicated mild TBI. All analyses were conducted using the total scores of the outcome instruments, except for the SF-36v2/-12v2, in which PCS and MCS were considered separately.

First, the Wei-Lachin analyses were carried out for all the instruments, including the GOSE/-Q, to obtain information about their sensitivity. For this purpose, six patient groups selected a priori (i.e., sex, age, education, premorbid psychological problems, clinical care pathways, and injury severity) nested in four TBI severity groups (uncomplicated and complicated mild TBI, moderate and severe TBI) were investigated. Second, the sensitivity of the PROMs was examined in relation to functional recovery. This approach was chosen to strengthen the evidence for the GOSE as a core measure in the field of TBI, to review the criticisms formulated regarding its applicability [15–18], and to consolidate the clinical relevance of the analyses in the present study. The analyses were performed for the patient groups nested in the following GOSE/-Q states, which differentiate between the three main recovery levels: severe disability (2/3-4), moderate disability (5–6), and good recovery (7–8) [12]. Since the cross-sectional sensitivity may vary for different time points of assessment, the analyses were performed using data collected at 3, 6, and 12 months after TBI. To identify the most sensitive instruments, we summarized the sensitivity of the instruments displaying at least medium effects in the pairwise group comparisons using percentages. The number of sensitive group comparisons varied from 0% (not sensitive to any group comparison) to 100% (sensitive to all group comparisons). The top three instruments displaying the highest sensitivity at each assessment point were identified. Finally, we provided recommendations for the selection of the most sensitive outcome instruments at the three time points after TBI. These were based on the effect sizes obtained from the Wei-Lachin analyses:

1. *Strongly recommended for use* (predominantly high sensitivity: MW effect size less than or equal to 0.29 or greater than or equal to 0.71)

2. *Recommended for use* (predominantly medium sensitivity: MW effect size beyond 0.36 or 0.64, but greater than 0.29 or less than 0.71)

3. *Little information gain* (predominantly small sensitivity: $0.36 \leq MW \leq 0.64$)

Only sensitive instruments can reliably determine the impairment in individual outcome domains. To provide clinicians and researchers with a further indicator for selecting the appropriate instrument for their purpose, we calculated the prevalence of impaired outcomes for each patient group at each time point. For the PROMs, impaired outcomes were determined using clinical cutoffs reported in previous studies (see description of instruments). For the GOSE/-Q, an outcome was considered impaired if recovery was rated as not complete (i.e., a GOSE/-Q score < 7). For an overview of the sensitivity analyses performed, see Fig 1.

**Missing data.** Two different approaches were considered for treating missing data: the analysis of patient data as available (i.e., at least one outcome assessment at one time point available) and the analysis of individuals with data available for all three time points (i.e., completers). We decided against imputing missing outcome data because the non-response rates were too high to perform imputation [89]. The results of the two approaches were compared, to determine the possible influence of the missing values. The effects for data as available were comparable with the data of participants who had completed all outcome measures at all three time points (completers' data). The analyses were therefore reported based on the data as available, as the higher number of cases leads to a higher test power. The completers' results are provided in the supplemental material.

Statistical analyses were performed using the TESTIMATE [90] software version V.6.5.14 for Wei-Lachin analyses and R version 4.0.2 [91] for descriptive statistics using the corrplot [92] package. The alpha level was adjusted for multiplicity using the Bonferroni correction depending on the type of analysis. For the analyses of sensitivity for all outcome instruments including the GOSE/-Q, the significance was set at $\alpha_{adj} = 0.00045$; for the group comparisons $\alpha_{adj} = 0.0001$ was applied. For comparisons of the PROMs in relation to the GOSE/-Q, $\alpha_{adj} = 0.005$ was used; for group comparisons $\alpha_{adj} = 0.0001$ was applied.

## Results

Depending on the outcome instrument and the time of the assessment, the sample size for the outcome assessments varied from N = 2088 (GAD-7) to N = 2842 (GOSE/Q) at 3 months, from N = 2181 (GAD-7) to N = 2760 (GOSE/-Q) at six months, and N = 1437 (SF-36v2) to N = 1977 (GOSE/-Q) at twelve months. Participants were predominately male (> 60%), younger than 65 years of age (approx. 75%) and had at least a secondary school certificate (approx. 70%). The majority reported having no premorbid psychological problems (> 50%). They had mainly suffered an uncomplicated (around 30%) or a complicated mild TBI (around 30%), followed by severe (10% to 19%) and moderate (5% to 8%) TBI. Patients were mostly admitted to an ICU (> 40%) and had an ISS > 10 (> 60%). Sample characteristics for each instrument and time point are shown in S2 Table. Fig 2 provides information on the sample sizes.

### Correlations between outcome domains

The outcome domains were moderately to highly correlated, except for the MCS and PCS of the SF-36v2/-12v2, which had a low correlation (< 0.30) with each other, justifying and requiring the use of multidimensional analyses. For details, see S1 Fig.

### Sensitivity of all outcome instruments

The GOSE/-Q displayed the highest sensitivity across all patient groups and time points. The PCS and MCS of the SF-36v2/-12v2, the QOLIBRI/-OS, and the RPQ were most sensitive in the group comparisons at one or more point in time (see S3 Table). For more details on the MW effect sizes, see S4 and S5 Tables.

### Sensitivity of the PROMs with respect to functional recovery status

**Overall sensitivity of the PROMs.** The overall sensitivity of the PROMs with respect to functional recovery was relatively stable at 3, 6, and 12 months after TBI, as determined by the average number of pairwise comparisons with an at least medium effect for the three time points. The PCS of the SF-36v2 and the QOLIBRI and their short forms distinguished best across all patient groups at all time points. Table 1 provides an overview on the overall

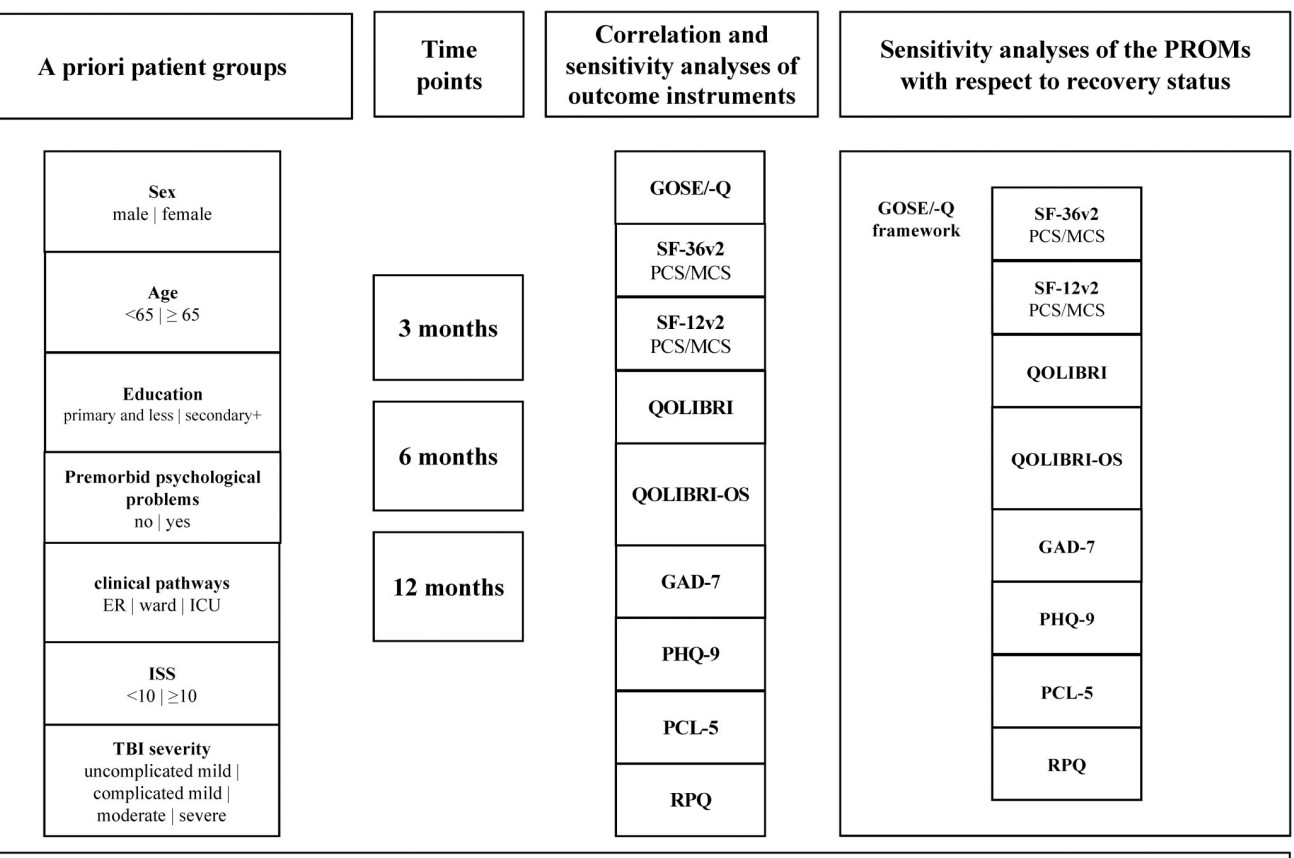

**Fig 1. Overview of sensitivity analyses.** ER = emergency room; ICU = intensive care unit; ISS = Injury Severity Scale; GOSE/-Q = Combined information on recovery status using the Glasgow Outcome Scale–Extended and its questionnaire version; SF-36v2 = 36-item Short Form Health Survey–version 2; SF-12v2 = 12-Item Short Form Survey–version 2; PCS = Physical Component Summary Score, MCS = Mental Component Summary Score; QOLIBRI = Quality of Life after Traumatic Brain Injury; QOLIBRI-OS = Quality of Life after Traumatic Brain Injury–Overall Scale; GAD-7 = Generalized Anxiety Disorder-7; PHQ-9 = Patient Health Questionnaire-9; PCL-5 = Posttraumatic Stress Disorder Checklist for DSM-5; RPQ = Rivermead Post-Concussion Symptoms Questionnaire.

sensitivity of the PROMs in the pairwise group comparisons regarding functional recovery status. For the sensitivity analyses of the data of the completers, see S6 Table. For more details on the MW effect sizes including instructions for interpretation, see S7 and S8 Tables. Some other instruments (e.g., RPQ, PHQ-9) showed differences in sensitivity in the six patient groups investigated. We will therefore further focus on the sensitivity of the PROMs with respect to six different patient groups and three time points after TBI with respect to the functional recovery status in greater detail.

**Sensitivity of the PROMs with respect to functional recovery status and sociodemographic factors.** Based on the stratification by sociodemographic characteristics (i.e., sex, age, education), the PCS of the SF-36v2/-12v2 and the QOLIBRI/-OS again demonstrated the highest ability to differentiate between good recovery and moderate/severe disability among all groups, with predominantly high effect sizes (i.e., less than or equal to 0.29 or greater than or equal to 0.71). Additionally, the RPQ displayed a high sensitivity in differentiating the recovery status within the group of the male patients at three months after TBI. The other PROMs demonstrated at least medium sensitivity at all time points, discriminating according to the functional recovery status and patient groups. The only exception was the GAD-7,

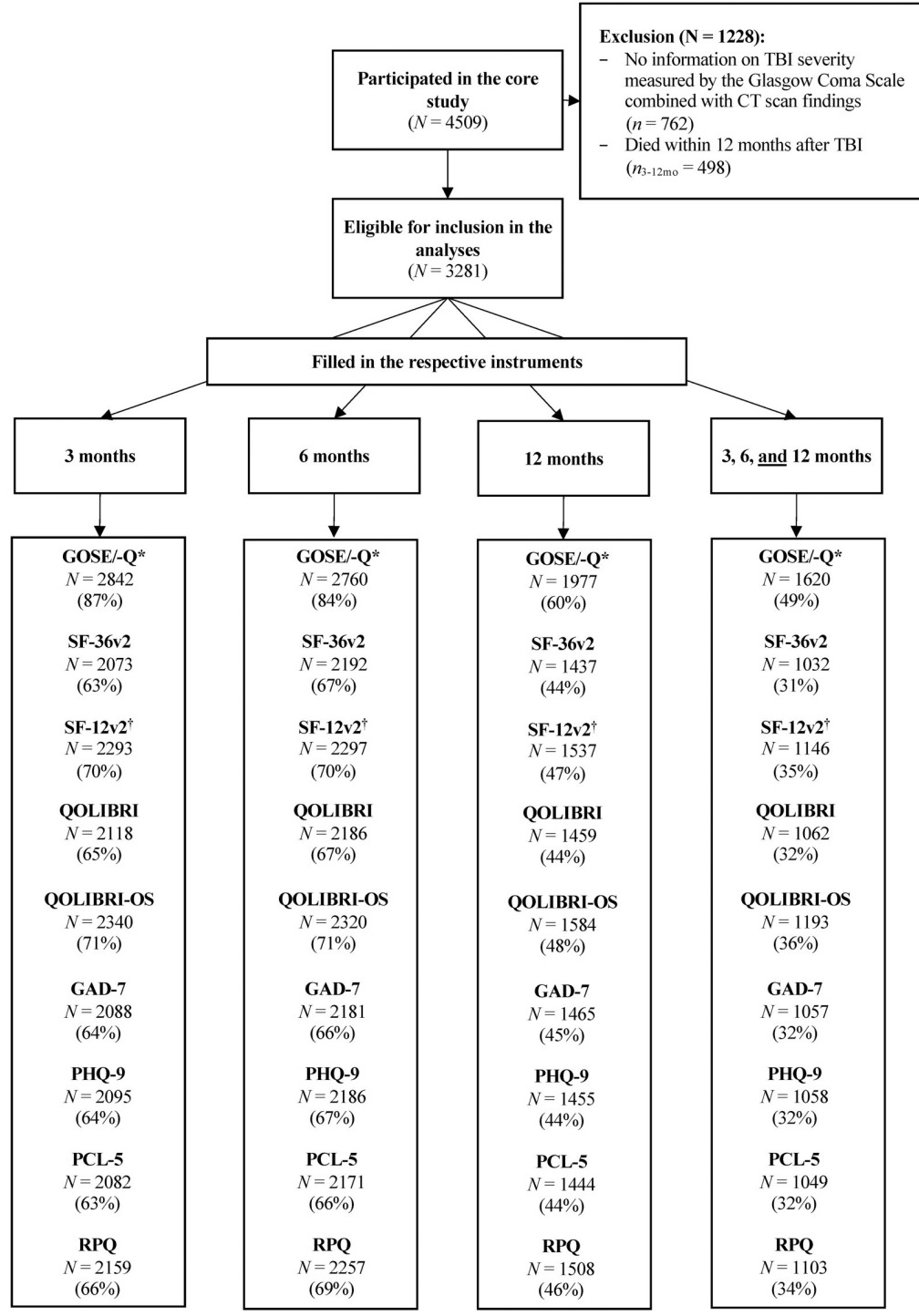

\* Missing GOSE values substituted by the GOSE-Q and/or clinical ratings

† Missing SF-12v2 values substituted by the values derived from the respective items of the SF-36v2

N = number, % = response rates.

**Fig 2. Numbers of completed instruments per time point (3, 6, and 12 months after TBI) and for all time points completed by the same individuals.** GOSE/-Q = Combined information on recovery status using the Glasgow Outcome Scale–Extended and its questionnaire version; SF-36v2 = 36-item Short Form Health Survey–version 2; SF-12v2 = 12-Item Short Form Survey–version 2; PCS = Physical Component Summary Score, MCS = Mental Component Summary Score; QOLIBRI = Quality of Life after Traumatic Brain Injury; QOLIBRI-OS = Quality of Life

after Traumatic Brain Injury–Overall Scale; GAD-7 = Generalized Anxiety Disorder-7; PHQ-9 = Patient Health
Questionnaire-9; PCL-5 = Posttraumatic Stress Disorder Checklist for DSM-5; RPQ = Rivermead Post-Concussion
Symptoms Questionnaire.

which displayed a low discriminative ability within the female group at three months, and the
age and primary education groups at twelve months.

**Sensitivity of the PROMs with respect to functional status and premorbid psychological
health status.** Concerning discriminating between individuals with and without premorbid
psychiatric problems with different recovery states, the PCS of the SF-36v2/-12v2 was again
the most sensitive at all time points, followed by the QOLIBRI/-OS. The GAD-7 and the MCS
of the SF-36v2/-12v2 were not able to discriminate well across all time points within the group
having premorbid psychiatric problems. All other PROMs displayed at least a medium and
thus satisfactory sensitivity across all patient groups and time points.

**Sensitivity of the PROMs with respect to functional status and injury-related factors.**
Inspecting the injury-related groups, the PCS of the SF-36v2/-12v2 and the QOLIBRI/-OS were
able to distinguish at all time points, followed by the RPQ, with medium to high MW effects. The
HRQOL measures, in particular, displayed high sensitivities in the comparison of TBI severity
groups and functional recovery status as well as of injury severity groups (ISS) and recovery status
at all time points. The RPQ was able to distinguish between good recovery and severe disability in
individuals after a moderate TBI as well as in those affected by moderate, severe, or profound inju-
ries (i.e., ISS < 10) at three months after TBI only. Additionally, the PHQ-9 was highly sensitive
to the functional recovery status in the group of individuals who were primarily admitted to the
ER and then discharged at three and six months after TBI. At twelve months, the PCL-5 was the
most sensitive instrument regarding all injury-related group comparisons. All other PROMs,
except for the GAD-7 and the SF-36v2/-12v2, revealed at least medium effects.

Fig 3 provides an example concerning which PROM can distinguish between good recovery
(GOSE 7–8) and moderate disability (GOSE 5–6) in complicated mild TBI, and to what extent.
Overall, the entire ensemble of PROMs displayed a high sensitivity in differentiating between
the patient groups selected a priori. The pooled effect was slightly above 0.29, CI95% [0.25,
0.33]. The group with good recovery displayed better outcomes compared with the group with

**Table 1. The overall sensitivity of the PROMs to pairwise group comparisons with respect to functional recovery status for three time points (patient data as
available).**

| Instrument | Three months | Six months | Twelve months | Average |
|---|---|---|---|---|
| | **n = 49** | **n = 45** | **n = 42** | |
| SF-36v2 PCS | **100%** | **96%** | 76% | **91%** |
| SF-12v2 PCS | **100%** | **93%** | 71% | **88%** |
| SF-36v2 MCS | 59% | 62% | 50% | 57% |
| SF-12v2 MCS | 61% | 60% | 55% | 59% |
| QOLIBRI | 92% | 71% | **90%** | **84%** |
| QOLIBRI-OS | **98%** | **73%** | 79% | 83% |
| GAD-7 | 57% | 60% | 43% | 53% |
| PHQ-9 | 67% | 69% | 64% | 67% |
| PCL-5 | 69% | 64% | 69% | 68% |
| RPQ | 69% | 69% | 67% | 68% |

Note. n = number of pairwise comparisons, % = percentage, average = average relative frequencies from 3 to 12 months. **Bold** values indicate the top three instruments
with the highest sensitivity in most group comparisons.

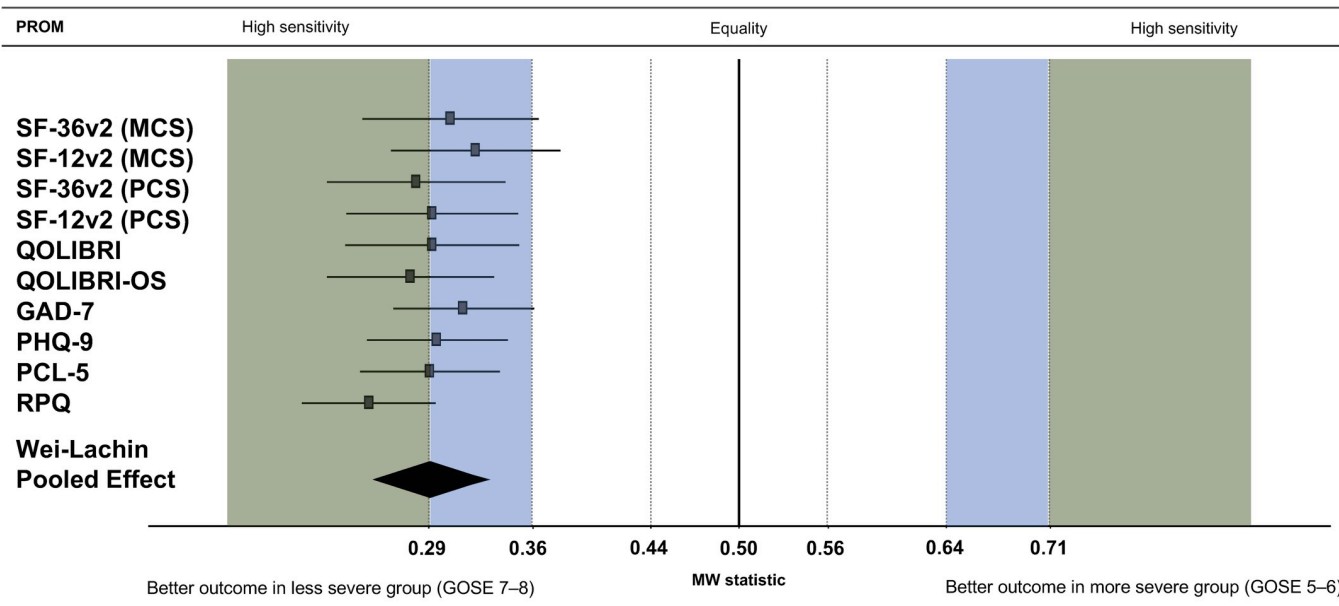

**Fig 3. Sensitivity of the PROMs in differentiating between good recovery (GOSE 7–8) and moderate disability (GOSE 5–6) for complicated mild TBI three months post-injury.** MW statistic: Mann-Whitney effect size with 95%-CI: confidence interval, Wei-Lachin pooled effect: pooled effect size combined across all PROMs for each group of interest. Green shaded area of the plot: large effect or high sensitivity (MW effect size less than or equal to 0.29 or greater than or equal to 0.71); blue shaded area: medium effect or medium sensitivity (MW effect size beyond 0.36 or 0.64, but greater than 0.29 or less than 0.71); transparent background: small effect or low sensitivity (0.36 ≤ MW ≤ 0.64).

the less favorable recovery, which was reflected by the high MW effect sizes. The RPQ, the PCS of the SF-36v2, and the QOLIBRI-OS presented the strongest effects and were thus most sensitive to the differences concerning the recovery status after complicated mild TBI. All other instruments had medium sensitivity, with CIs not exceeding the cut-off of 0.36. This indicated that the effect was stable medium with a 95% probability. An exception was the MCS of both forms of the SF instruments, where the lower CI cut-off was in the low sensitivity range (i.e., below 0.36). For details concerning the pooled (i.e., combined) MW effect sizes of the PROMs and respective effects in different groups of interest, see S1 Text.

**Recommendations for the selection of the most sensitive PROMs with respect to functional status and six different patient groups at 3, 6, and 12 months after TBI.** Most PROMs displayed a high to medium sensitivity with respect to the recovery status, across all the investigated patient groups and at all time points. However, the MCS of the SF-36v2/-12v2 and the GAD-7 did not discriminate well in some patient groups (e.g., sex, age, premorbid health status). Figs 4–6 summarize these recommendations.

Based on the overall sensitivity of the PROMs as well as the findings of the patient-group and time point analyses, the PCS of the SF-36v2/12v2, the QOLIBRI/-OS, extended by an assessment of post-concussion symptoms using the RPQ, can be recommended to complement information on the recovery status. In addition, an assessment of depression using the PHQ-9 would provide additional information on the psychological state, especially in those discharged from the ER but showing less favorable functional recovery.

## Prevalence of impaired outcomes

Because all instruments were moderately to highly sensitive with respect to all patient groups, the prevalence of impaired outcomes could be reliably calculated to provide additional

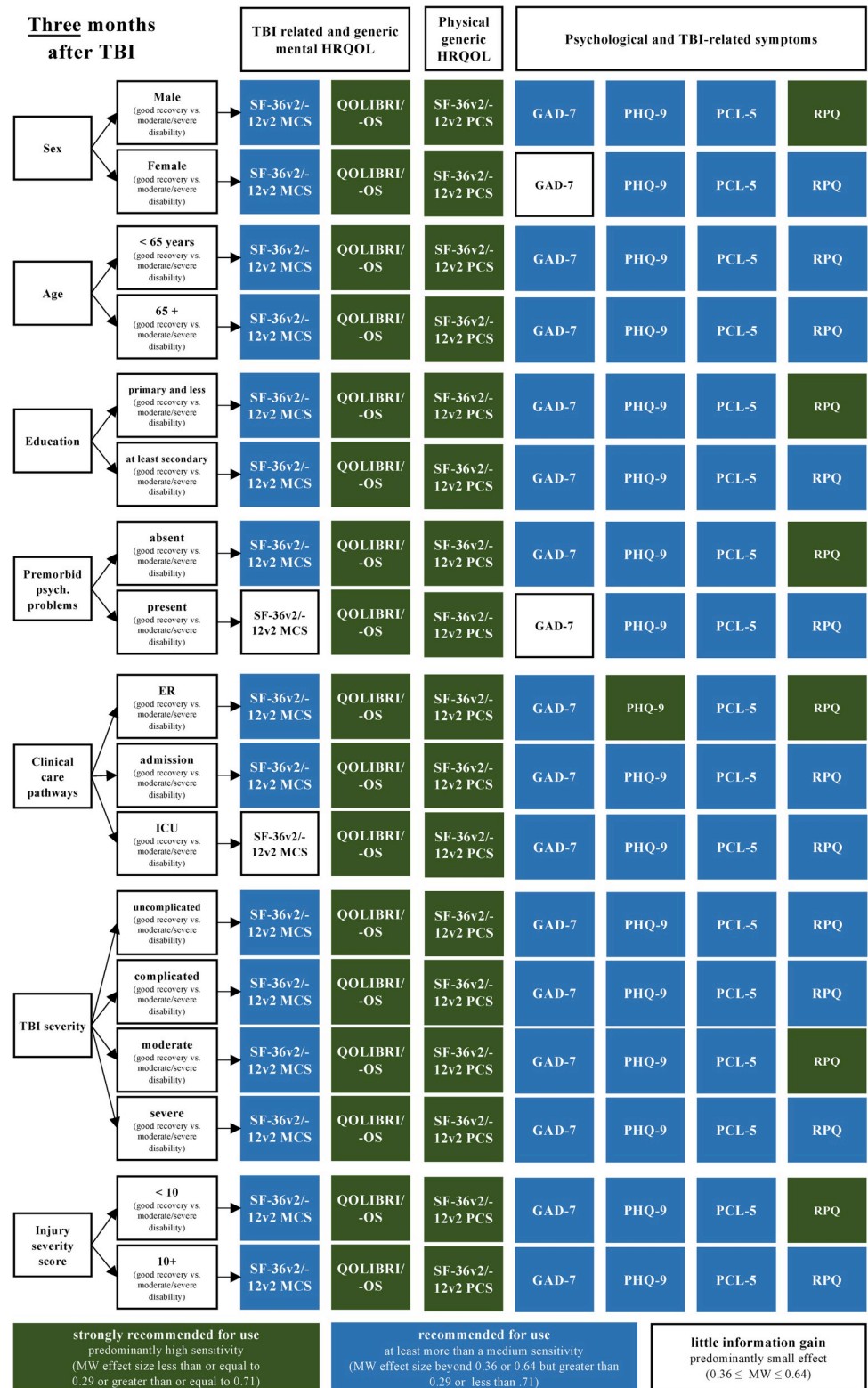

**Fig 4. PROMs recommended for use with respect to different sociodemographic, premorbid, and injury-related patient groups and recovery statuses (i.e., GOSE/-Q as reference) at three months after TBI based on the MW effect sizes.** Numbers are documented in S7 Table (data as available) and in S8 Table (completers).

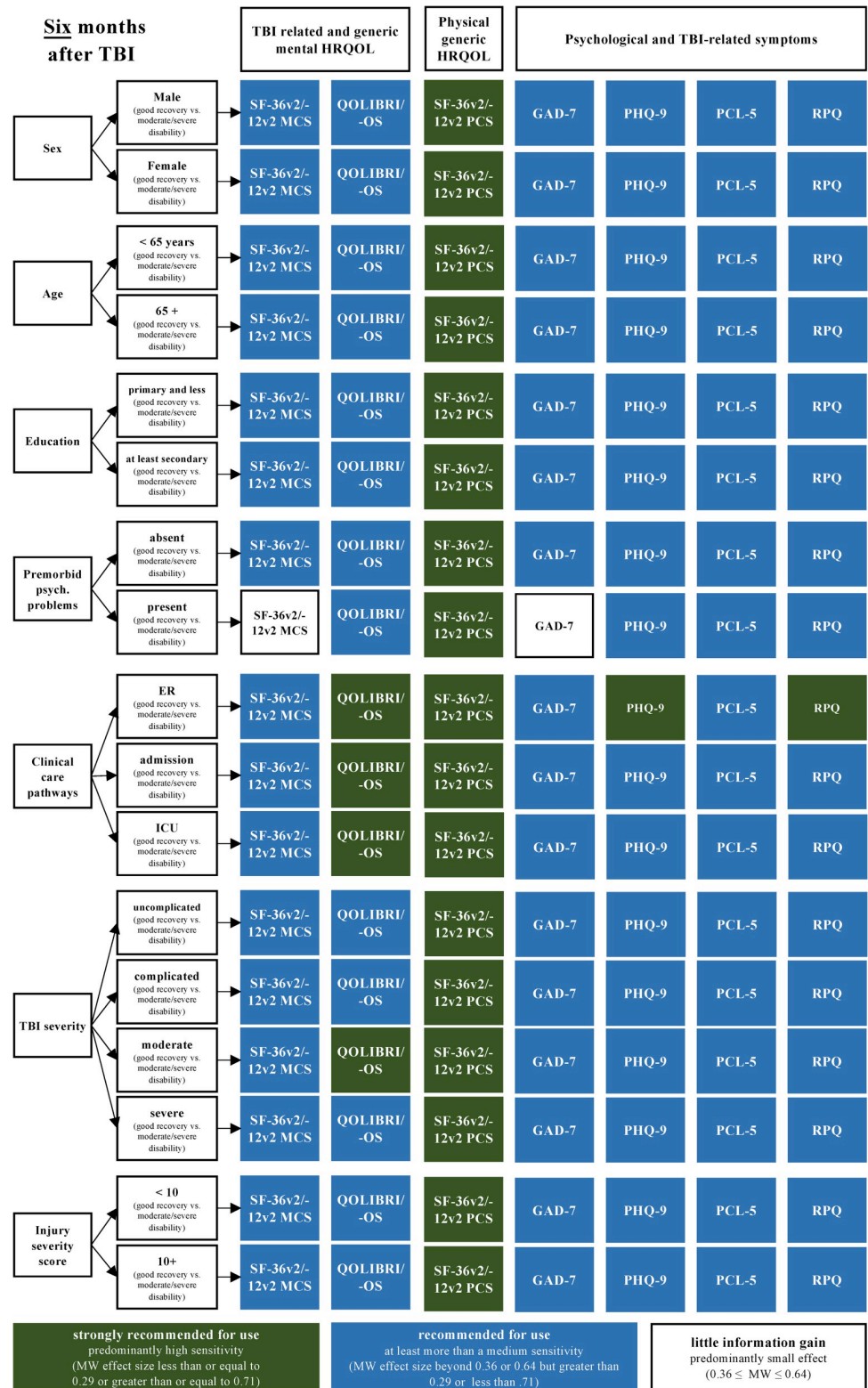

**Fig 5. PROMs recommended for use with respect to different sociodemographic, premorbid, and injury-related patient groups and recovery statuses (i.e., GOSE/-Q as reference) at six months after TBI based on the MW effect sizes.** Numbers are documented in S7 Table (data as available) and S8 Table (completers).

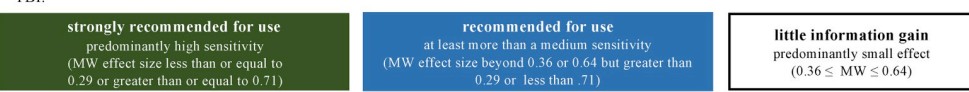

**Twelve months after TBI**

| | | TBI related and generic mental HRQOL | | Physical generic HRQOL | Psychological and TBI-related symptoms | | | |
|---|---|---|---|---|---|---|---|---|
| **Sex** | **Male** (good recovery vs. moderate/severe disability) | SF-36v2/-12v2 MCS | QOLIBRI/-OS | SF-36v2/-12v2 PCS | GAD-7 | PHQ-9 | PCL-5 | RPQ |
| | **Female** (good recovery vs. moderate/severe disability) | SF-36v2/-12v2 MCS | QOLIBRI/-OS | SF-36v2/-12v2 PCS | GAD-7 | PHQ-9 | PCL-5 | RPQ |
| **Age** | **< 65 years** (good recovery vs. moderate/severe disability) | SF-36v2/-12v2 MCS | QOLIBRI/-OS | SF-36v2/-12v2 PCS | GAD-7 | PHQ-9 | PCL-5 | RPQ |
| | **65 +** (good recovery vs. moderate/severe disability) | SF-36v2/-12v2 MCS | QOLIBRI/-OS | SF-36v2/-12v2 PCS | GAD-7 | PHQ-9 | PCL-5 | RPQ |
| **Education** | **primary and less** (good recovery vs. moderate/severe disability) | SF-36v2/-12v2 MCS | QOLIBRI/-OS | SF-36v2/-12v2 PCS | GAD-7 | PHQ-9 | PCL-5 | RPQ |
| | **at least secondary** (good recovery vs. moderate/severe disability) | SF-36v2/-12v2 MCS | QOLIBRI/-OS | SF-36v2/-12v2 PCS | GAD-7 | PHQ-9 | PCL-5 | RPQ |
| **Premorbid psych. problems** | **absent** (good recovery vs. moderate/severe disability) | SF-36v2/-12v2 MCS | QOLIBRI/-OS | SF-36v2/-12v2 PCS | GAD-7 | PHQ-9 | PCL-5 | RPQ |
| | **present** (good recovery vs. moderate/severe disability) | SF-36v2/-12v2 MCS | QOLIBRI/-OS | SF-36v2/-12v2 PCS | GAD-7 | PHQ-9 | PCL-5 | RPQ |
| **Clinical care pathways[1]** | **admission** (good recovery vs. moderate/severe disability) | SF-36v2/-12v2 MCS | QOLIBRI/-OS | SF-36v2/-12v2 PCS | GAD-7 | PHQ-9 | PCL-5 | RPQ |
| | **ICU** (good recovery vs. moderate/severe disability) | SF-36v2/-12v2 MCS | QOLIBRI/-OS | SF-36v2/-12v2 PCS | GAD-7 | PHQ-9 | PCL-5 | RPQ |
| **TBI severity[2]** | **uncomplicated** (good recovery vs. moderate/severe disability) | SF-36v2/-12v2 MCS | QOLIBRI/-OS | SF-36v2/-12v2 PCS | GAD-7 | PHQ-9 | PCL-5 | RPQ |
| | **complicated** (good recovery vs. moderate/severe disability) | SF-36v2/-12v2 MCS | QOLIBRI/-OS | SF-36v2/-12v2 PCS | GAD-7 | PHQ-9 | PCL-5 | RPQ |
| | **severe** (good recovery vs. moderate/severe disability) | SF-36v2/-12v2 MCS | QOLIBRI/-OS | SF-36v2/-12v2 PCS | GAD-7 | PHQ-9 | PCL-5 | RPQ |
| **Injury severity score** | **< 10** (good recovery vs. moderate/severe disability) | SF-36v2/-12v2 MCS | QOLIBRI/-OS | SF-36v2/-12v2 PCS | GAD-7 | PHQ-9 | PCL-5 | RPQ |
| | **10+** (good recovery vs. moderate/severe disability) | SF-36v2/-12v2 MCS | QOLIBRI/-OS | SF-36v2/-12v2 PCS | GAD-7 | PHQ-9 | PCL-5 | RPQ |

[1] Based on the study design, participants seen in the emergency room (ER) and then discharged were not included in the 12-months follow up assessments.
[2] Due to a low number of participants ($n \leq 29$), no information is available on sensitivity of the instruments for the individuals after moderate TBI.

| **strongly recommended for use** predominantly high sensitivity (MW effect size less than or equal to 0.29 or greater than or equal to 0.71) | **recommended for use** at least more than a medium sensitivity (MW effect size beyond 0.36 or 0.64 but greater than 0.29 or less than .71) | **little information gain** predominantly small effect ($0.36 \leq$ MW $\leq 0.64$) |
|---|---|---|

**Fig 6. PROMs recommended for use with respect to different sociodemographic, premorbid, and injury-related patient groups and recovery statuses (i.e., GOSE/-Q as reference) at six months after TBI based on the MW effect sizes.** Numbers are documented in S7 Table (data as available) and in S8 Table (completers).

information for clinicians and researchers (for details, see S9 Table). Impairment varied across the instruments and time points. The functional recovery status was impaired in 41% to 46% of the patients, followed by clinically relevant post-concussion symptoms (37% to 38%), and impaired physical generic HRQOL as measured by the PCS of the SF-12v2 (27% to 36%) from three to twelve months. Overall, 22% to 26% reported impaired TBI-specific HRQOL, with a slightly higher number of impaired individuals at three months. From three to twelve months after TBI, the number of individuals with unfavorable outcomes decreased. Overall, the prevalence of impaired outcomes varied from 10% (depression and PTSD) to 41% (functional recovery status).

Individuals with the following sociodemographic, premorbid, and injury-related characteristics were more likely to show impaired outcomes across all instruments and time points: being male, younger than 65 years of age, having graduated from at least secondary school, reporting no premorbid psychological problems, being admitted to ICU, having had an ISS of 10 and above, and having suffered either a mild or severe TBI. Individuals after mild TBI were most impaired in all outcome domains except for functional recovery, whereas individuals after severe TBI more frequently displayed an unfavorable recovery but reported no impairments in other domains. Individuals after uncomplicated mild TBI were impaired slightly more frequently regarding certain psychological outcomes, such as anxiety and PTSD, compared to those after a complicated mild TBI at three and six months. In contrast, individuals after a complicated mild TBI in general more often experienced impaired generic and TBI-specific HRQOL.

## Discussion

The present study aimed to analyze the multidimensional cross-sectional sensitivity of the outcome instruments commonly used in the field of TBI. The study results present evidence for the sensitivity of these instruments to different clinically relevant patient groups selected a priori, at three time points after TBI. Thus, a general recommendation has been given for selecting appropriate measures for clinicians and researchers for administration within one year after injury as well as for three different time points post TBI.

The GOSE/-Q showed the highest discriminatory ability. Therefore, in line with other studies [93, 94] and contradicting some of the critical findings [15–18], its administration as a clinical standard in the field of TBI and its use as a reference for further sensitivity analyses was supported.

Recent studies [93, 95, 96] suggest that the use of a single outcome instrument, such as the GOSE, may not provide a comprehensive picture of patients' health status. For a comprehensive representation of a patient's clinical picture, it is recommended that multidimensional outcome assessments are performed. The medium to high sensitivity of the PROMs to the recovery status of the patients supports the approach of complementing the sole GOSE assessment by other clinically relevant outcome measures. Based on the results of our analyses, the use of the physical generic HRQOL component of the SF-36v2/-12v2 instruments (PCS), the disease-specific HRQOL measures (QOLIBRI/-OS) and the RPQ to assess post-concussion symptoms can be recommended to provide a sensitive and valid multidimensional assessment of outcomes and impairments in combination with the GOSE.

The short forms of the HRQOL instruments (i.e., QOLIBRI/-OS measuring TBI-specific and SF-36v2/-12v2 measuring generic HRQOL) showed comparable sensitivity to their longer versions. Therefore, they could be useful in both routine clinical assessment and research to reduce patient burden and to save assessment time. If recovery status is not assessed, the results of the first analytic approach of sensitivity (i.e., irrespective of recovery status) can be used to select the most sensitive outcome instruments.

As the sensitivity of the outcome instruments differed slightly at the three time points, it will now be discussed in greater detail along the timeline of one year post TBI.

At **three months,** PROMs assessing generic physical and TBI-specific HRQOL, and post-concussion(-like) symptoms are most sensitive in detecting differences in the recovery states, followed by PROMs assessing depression. Patients with moderate/severe disability tend to report lower HRQOL, more intense post-concussion symptoms, and severe major depression symptoms. Similar findings have already been reported based on univariate analyses [55, 95, 97]. Individuals discharged from the ER still recover poorly and experience more severe depressive symptoms, which is reflected by the MW effect sizes. These findings indicate a possible undertreatment of those discharged from the ER. For example, Ganti and colleagues (2015) [98] found that 5% of individuals discharged after a mild TBI return to the ER within 72 hours. Especially those with CT abnormalities are at risk of developing post-concussion symptoms and pain needing further treatment. Additionally, as in other studies [4], we observed a relatively high prevalence (>30%) of clinically relevant post-concussion symptoms in this study, especially in individuals after (complicated) mild TBI. Since depression post TBI has a significant impact on health, work participation, social relationships, and HRQOL in all TBI severity groups [99, 100], it should be properly clinically diagnosed and treated early on during the clinical care pathway, as well as in outpatient care [101]. If time and patient burden allow and there is no clinical assessment of depression, the severity of major depression should be assessed using the PHQ-9 (also longitudinally) even though only a medium sensitivity was observed.

Not only the TBI itself, but also other injuries related to the cause of the trauma can affect the outcome and recovery of the patients [40, 102]. It is therefore important to monitor the status of trauma severity, as measured by the ISS. In our study, three months after TBI, lower ISS and poorer recovery status were associated with worse outcomes, particularly concerning generic and disease-specific HRQOL and post-concussion(-like) symptoms. We therefore recommend the use of SF-36v2/12v2, QOLIBRI/-OS, and the RPQ also in patients after extracranial injuries and TBI to assess these domains and determine intervention needs.

At **six months** after TBI, especially the physical generic HRQOL component of the SF-36v2/12v2 instruments displayed a high sensitivity in distinguishing within and between the different patient groups. The TBI-specific HRQOL measures, in particular, are highly sensitive when it comes to detecting patients following different clinical care pathways and those after a moderate TBI. These instruments have a medium discriminative ability across the other patient groups, which might be attributable to the fact that functional recovery is relatively strongly associated with the physical HRQOL component of the SF-36v2/12v2 measures. Thus, those individuals who recover well are more likely to report a higher physical generic HRQOL than those who are still experiencing functional problems. Additionally, individuals who were only treated as outpatients (i.e., in the ER) and recovered less well still seem to suffer more from depression and post-concussion symptoms compared to those who made a good functional recovery. This again indicates that follow-up screenings during the hospital stay, as well as later during outpatient care, are of great importance so as to detect symptom manifestation or aggravation, to provide appropriate treatment, and to facilitate the recovery process. In addition, particular attention should be paid to patients with extracranial injuries and major trauma, and to their HRQOL [103].

At **twelve months**, the HRQOL instruments were especially sensitive in detecting changes in the recovery status. However, the GAD-7 assessing generalized anxiety disorders was less sensitive in distinguishing between the different patient groups, whereas the PHQ-9 assessing depression displayed stable medium sensitivity. The prevalence of major depression in our study was around 10% in the total sample, whereas in a meta-analysis of 99 studies [104] it was

38%. The authors of that study reported a high association between mild TBI and depressive symptoms for three time points. If we consider the TBI severity when calculating prevalence, we obtain comparable results. Around 13% of patients after an uncomplicated and 36% after a complicated mild TBI report clinically relevant depression. Taken together, these findings underline that an adequate early and longitudinal evaluation using a sensitive measure allows depression to be treated, facilitating a successful return to everyday life. The PCL-5 assessing PTSD was moderately sensitive. It captured an increase in the inpatient group admitted to a hospital ward at twelve months compared to the three- and six-month assessments. These findings are in line with a recent meta-analysis involving 52 studies [24] in which the authors reported that PTSD, if persistent, remains high a long time (i.e., up to five years) after the TBI and shows no clear decrease.

Summarizing the detailed information on the sensitivity of the instruments at the three points in time after TBI, the instruments are best at discriminating between and within all patient groups with reference to functional recovery at three months. This can potentially be attributed to the fact that the symptom burden is most prominent at this time point. At six months, however, the impairments decrease slightly, as the symptom burden may fluctuate, whereas at twelve months the negative impact of the TBI may have chronified but remains lower compared to three months after TBI.

To further develop post-TBI care, treatment, and rehabilitation, an assessment of potential deficits should be conducted as early as possible and longitudinally using reliable, valid, and sensitive instruments that measure the consequences of the TBI multidimensionally in all relevant health states and life domains. The PROMs analyzed in the present study provide this basis. The use of these instruments in combination with the GOSE would again allow timely diagnosis and treatment at follow-up visits, which should be performed at several time points at least up to one year after TBI to help to control, prevent, or reduce the manifestation of symptoms in various outcome domains.

## Limitations

Despite a relatively high total sample size, not all pairwise comparisons could be carried out because of the small number of cases within certain patient groups. Therefore, the significance of small effects could be compromised by lower test power, and the generalizability of the results may be limited. We are aware that the Bonferroni correction, which was used to avoid alpha-error inflation in multiple group comparisons, is a conservative adjustment method associated with diminished test power [105]. However, it allows group comparisons to become significant with a low probability of error, making our results more stable. Minor differences in sensitivity at different time points may be attributed to differences in the number of participants at the 3-, 6-, and 12-month outcome assessments. Nonetheless, the analyses of the complete data suggest a stable sensitivity of the PROMs, even if the number of observations available for all time points is reduced.

Due to the general design of the CENTER-TBI study, some groups within the care pathway were not involved in all follow-up assessments (i.e., the ER group at twelve months). A further investigation at later time points (i.e., beyond six and twelve months) could therefore provide helpful insights into the longitudinal development of outcomes in individuals after TBI, especially those discharged from the ER.

In the present study, the investigation of some areas affected by TBI may be underrepresented. First, our sample consisted predominantly of individuals after mild TBI. Therefore, transferring the recommendations concerning the selection of outcome instruments to moderate and severe TBI should be done with caution. Second, we lacked data concerning work

participation and return to daily life, for example, as well as family and caregiver burden [7, 8]. In addition, some protective factors, such as resilience [106] and a stable social and economic environment, as well as social participation [107, 108], could be included in future research to provide more insight into a multidimensional longitudinal development of outcomes after TBI. In our study, the information regarding psychiatric problems before and after the TBI of the participants was based solely on self-reported data. Standardized clinical diagnoses of depression, anxiety, and PTSD as well as information on psychopharmacological treatment effects would contribute to a more precise differentiation, providing valuable directions for future studies. The sensitivity to detecting drug effects can however only be evaluated once the sensitivity of instruments to relevant predictors or risk factors has already been established. Furthermore, the functional recovery status of the study participants was determined using the GOSE, with missing values substituted based on the GOSE-Q and/or clinical assessments. Therefore, it is not possible to compare the sensitivity of the GOSE interview with its question-naire version. In a recent study, GOSE ratings showed good agreement with GOSE-Q scores and a similar association with other outcomes after TBI [109]. However, future studies should further address the sensitivity of the two GOSE forms to provide more evidence for their appli-cability and mutual substitution. Finally, the sole use of the PCS of the SF-36v2/-12v2, which has shown the highest discriminatory ability among other PROMs with respect to functional recovery, should be further validated in the field of TBI.

To gain better insight into the course of recovery after a TBI, future research should exam-ine how multiple psychological and symptom-related PROMs are associated with trajectories of the functional recovery status.

## Conclusion

The present study provides the first systematic multidimensional sensitivity analyses of out-come instruments at three time points within one year after TBI using a literature-based a pri-ori selection of groups with and without reference to recovery status. For a sensitive, reliable, economic, yet comprehensive assessment of outcomes after TBI, the evaluation of the recovery status should be combined with self-reports on physical generic HRQOL (e.g., PCS of the SF-12v2), disease-specific HRQOL (e.g., QOLIBRI-OS), and post-concussion symptoms (RPQ). If time and patient burden allow, the severity of major depression should additionally be assessed with the PHQ-9 if it was not diagnosed clinically. The suggested, relatively short multidimen-sional yet comprehensive outcome assessment of individuals after TBI of all severities may help to evaluate treatment effects sensitively and tailor interventions and care after TBI.

## Supporting information

**S1 Table. Overview of studies on protective and risk factors for the selected outcome areas after TBI.**
(PDF)

**S2 Table. Sample characteristics per outcome instrument and time point.**
(PDF)

**S3 Table. Sensitivity of the outcome instruments to the pairwise group comparisons.**
(PDF)

**S4 Table. Sensitivity of all outcome instruments to different patient groups (data as avail-able).**
(PDF)

**S5 Table. Sensitivity of all outcome instruments to different patient groups (completers data).**
(PDF)

**S6 Table. Sensitivity of the PROMs with respect to functional recovery and pairwise group comparisons (completers' data).**
(PDF)

**S7 Table. Sensitivity of the PROMs with respect to functional recovery and pairwise group comparisons (data as available).**
(PDF)

**S8 Table. Sensitivity of the PROMs with respect to functional recovery and pairwise group comparisons (completers' data).**
(PDF)

**S9 Table. Number of individuals with impaired outcomes with respect to instruments' cut-off values at three, six, and twelve months after TBI stratified by sociodemographic, pre-morbid, and injury-related factors.**
(PDF)

**S1 Text. Sensitivity of the outcome instruments in respect to the recovery status (forest plots).**
(PDF)

**S1 Fig. Correlations between outcomes.** Graded blue ellipses indicate low (light blue) to high (dark blue) negative correlations, while graded green ellipses indicate low (light green) to high (dark green) positive correlations. GOSE/-Q = Combined information on recovery status using the Glasgow Outcome Scale–Extended and its questionnaire version; SF-36v2 = 36-item Short Form Health Survey–version 2; SF-12v2 = 12-Item Short Form Survey–version 2; PCS = Physical Component Summary Score, MCS = Mental Component Summary Score; QOLIBRI = Quality of Life after Traumatic Brain Injury; QOLIBRI-OS = Quality of Life after Traumatic Brain Injury–Overall Scale; GAD-7 = Generalized Anxiety Disorder-7; PHQ-9 = Patient Health Questionnaire-9; PCL-5 = Posttraumatic Stress Disorder Checklist for DSM-5; RPQ = Rivermead Post-Concussion Symptoms Questionnaire.
(TIF)

## Acknowledgments

We gratefully thank all CENTER-TBI participants and investigators (lead author: Andrew I.R. Maas, andrew.maas@uza.be):

Cecilia Åkerlund[1], Krisztina Amrein[2], Nada Andelic[3], Lasse Andreassen[4], Audny Anke[5], Anna Antoni[6], Gérard Audibert[7], Philippe Azouvi[8], Maria Luisa Azzolini[9], Ronald Bartels[10], Pál Barzó[11], Romuald Beauvais[12], Ronny Beer[13], Bo-Michael Bellander[14], Antonio Belli[15], Habib Benali[16], Maurizio Berardino[17], Luigi Beretta[9], Morten Blaabjerg[18], Peter Bragge[19], Alexandra Brazinova[20], Vibeke Brinck[21], Joanne Brooker[22], Camilla Brorsson[23], Andras Buki[24], Monika Bullinger[25], Manuel Cabeleira[26], Alessio Caccioppola[27], Emiliana Calappi [27], Maria Rosa Calvi[9], Peter Cameron[28], Guillermo Carbayo Lozano[29], Marco Carbonara[27], Simona Cavallo[17], Giorgio Chevallard[30], Arturo Chieregato[30], Giuseppe Citerio[31, 32], Hans Clusmann[33], Mark Coburn[34], Jonathan Coles[35], Jamie D. Cooper[36], Marta Correia[37], Amra Čović [38], Nicola Curry[39], Endre Czeiter[24], Marek Czosnyka[26], Claire Dahyot-Fizelier[40], Paul Dark[41], Helen Dawes[42], Véronique De Keyser[43], Vincent Degos[16], Francesco Della Corte[44],

Hugo den Boogert[10], Bart Depreitere[45], Đula Đilvesi [46], Abhishek Dixit[47], Emma Donoghue[22], Jens Dreier[48], Guy-Loup Dulière[49], Ari Ercole[47], Patrick Esser[42], Erzsébet Ezer[50], Martin Fabricius[51], Valery L. Feigin[52], Kelly Foks[53], Shirin Frisvold[54], Alex Furmanov[55], Pablo Gagliardo[56], Damien Galanaud[16], Dashiell Gantner[28], Guoyi Gao[57], Pradeep George[58], Alexandre Ghuysen[59], Lelde Giga[60], Ben Glocker[61], Jagoš Golubovic[46], Pedro A. Gomez [62], Johannes Gratz[63], Benjamin Gravesteijn[64], Francesca Grossi[44], Russell L. Gruen[65], Deepak Gupta[66], Juanita A. Haagsma[64], Iain Haitsma[67], Raimund Helbok[13], Eirik Helseth[68], Lindsay Horton [69], Jilske Huijben[64], Peter J. Hutchinson[70], Bram Jacobs[71], Stefan Jankowski[72], Mike Jarrett[21], Ji-yao Jiang[58], Faye Johnson[73], Kelly Jones[52], Mladen Karan[46], Angelos G. Kolias[70], Erwin Kompanje[74], Daniel Kondziella[51], Evgenios Kornaropoulos[47], Lars-Owe Koskinen[75], Noémi Kovács[76], Ana Kowark[77], Alfonso Lagares[62], Linda Lanyon[58], Steven Laureys[78], Fiona Lecky[79, 80], Didier Ledoux[78], Rolf Lefering[81], Valerie Legrand[82], Aurelie Lejeune[83], Leon Levi[84], Roger Lightfoot[85], Hester Lingsma[64], Andrew I.R. Maas[43], Ana M. Castaño-León[62], Marc Maegele[86], Marek Majdan[20], Alex Manara[87], Geoffrey Manley[88], Costanza Martino[89], Hugues Maréchal[49], Julia Mattern[90], Catherine McMahon[91], Béla Melegh[92], David Menon[47], Tomas Menovsky[43], Ana Mikolic[64], Benoit Misset[78], Visakh Muraleedharan[58], Lynnette Murray[28], Ancuta Negru[93], David Nelson[1], Virginia Newcombe[47], Daan Nieboer[64], József Nyirádi[2], Otesile Olubukola[79], Matej Oresic[94], Fabrizio Ortolano[27], Aarno Palotie[95, 96, 97], Paul M. Parizel[98], Jean-François Payen[99], Natascha Perera[12], Vincent Perlbarg[16], Paolo Persona[100], Wilco Peul[101], Anna Piippo-Karjalainen[102], Matti Pirinen[95], Dana Pisica[64], Horia Ples[93], Suzanne Polinder[64], Inigo Pomposo[29], Jussi P. Posti [103], Louis Puybasset[104], Andreea Radoi [105], Arminas Ragauskas[106], Rahul Raj[102], Malinka Rambadagalla[107], Isabel Retel Helmrich[64], Jonathan Rhodes[108], Sylvia Richardson[109], Sophie Richter[47], Samuli Ripatti[95], Saulius Rocka[106], Cecilie Roe[110], Olav Roise[111,112], Jonathan Rosand[113], Jeffrey V. Rosenfeld[114], Christina Rosenlund[115], Guy Rosenthal[55], Rolf Rossaint[77], Sandra Rossi[100], Daniel Rueckert[61] Martin Rusnák[116], Juan Sahuquillo[105], Oliver Sakowitz[90, 117], Renan Sanchez-Porras[117], Janos Sandor[118], Nadine Schäfer[81], Silke Schmidt[119], Herbert Schoechl[120], Guus Schoonman[121], Rico Frederik Schou[122], Elisabeth Schwendenwein[6], Charlie Sewalt[64], Ranjit D. Singh[101], Toril Skandsen[123, 124], Peter Smielewski[26], Abayomi Sorinola[125], Emmanuel Stamatakis[47], Simon Stanworth[39], Robert Stevens[126], William Stewart[127], Ewout W. Steyerberg[64, 128], Nino Stocchetti[129], Nina Sundström[130], Riikka Takala[131], Viktória Tamás[125], Tomas Tamosuitis[132], Mark Steven Taylor[20], Braden Te Ao[52], Olli Tenovuo[103], Alice Theadom[52], Matt Thomas[87], Dick Tibboel[133], Marjolein Timmers[74], Christos Tolias[134], Tony Trapani[28], Cristina Maria Tudora[93], Andreas Unterberg[90], Peter Vajkoczy[135], Shirley Vallance[28], Egils Valeinis[60], Zoltán Vámos[50], Mathieu van der Jagt[136], Gregory Van der Steen[43], Joukje van der Naalt[71], Jeroen T.J.M. van Dijck [101], Inge A. van Erp[101], Thomas A. van Essen[101], Wim Van Hecke[137], Caroline van Heugten[138], Dominique Van Praag[139], Ernest van Veen[64], Thijs Vande Vyvere[137], Roel P. J. van Wijk[101], Alessia Vargiolu[32], Emmanuel Vega[83], Kimberley Velt[64], Jan Verheyden[137], Paul M. Vespa[140], Anne Vik[123, 141], Rimantas Vilcinis[132], Victor Volovici[67], Nicole von Steinbüchel[38], Daphne Voormolen[64], Petar Vulekovic[46], Kevin K.W. Wang[142], Daniel Whitehouse[47], Eveline Wiegers[64], Guy Williams[47], Lindsay Wilson[69], Stefan Winzeck[47], Stefan Wolf[143], Zhihui Yang[113], Peter Ylén[144], Alexander Younsi[90], Frederick A. Zeiler[47,145], Veronika Zelinkova[20], Agate Ziverte[60], Tommaso Zoerle[27]

1 Department of Physiology and Pharmacology, Section of Perioperative Medicine and Intensive Care, Karolinska Institutet, Stockholm, Sweden

2 János Szentágothai Research Centre, University of Pécs, Pécs, Hungary

3 Division of Surgery and Clinical Neuroscience, Department of Physical Medicine and Rehabilitation, Oslo University Hospital and University of Oslo, Oslo, Norway

4 Department of Neurosurgery, University Hospital Northern Norway, Tromso, Norway

[5] Department of Physical Medicine and Rehabilitation, University Hospital Northern Norway, Tromso, Norway

[6] Trauma Surgery, Medical University Vienna, Vienna, Austria

[7] Department of Anesthesiology & Intensive Care, University Hospital Nancy, Nancy, France

[8] Raymond Poincare hospital, Assistance Publique–Hopitaux de Paris, Paris, France

[9] Department of Anesthesiology & Intensive Care, S Raffaele University Hospital, Milan, Italy

[10] Department of Neurosurgery, Radboud University Medical Center, Nijmegen, The Netherlands

[11] Department of Neurosurgery, University of Szeged, Szeged, Hungary

[12] International Projects Management, ARTTIC, Munchen, Germany

[13] Department of Neurology, Neurological Intensive Care Unit, Medical University of Innsbruck, Innsbruck, Austria

[14] Department of Neurosurgery & Anesthesia & intensive care medicine, Karolinska University Hospital, Stockholm, Sweden

[15] NIHR Surgical Reconstruction and Microbiology Research Centre, Birmingham, UK

[16] Anesthesie-Réanimation, Assistance Publique–Hopitaux de Paris, Paris, France

[17] Department of Anesthesia & ICU, AOU Città della Salute e della Scienza di Torino—Orthopedic and Trauma Center, Torino, Italy

[18] Department of Neurology, Odense University Hospital, Odense, Denmark

[19] BehaviourWorks Australia, Monash Sustainability Institute, Monash University, Victoria, Australia

[20] Department of Public Health, Faculty of Health Sciences and Social Work, Trnava University, Trnava, Slovakia

[21] Quesgen Systems Inc., Burlingame, California, USA

[22] Australian & New Zealand Intensive Care Research Centre, Department of Epidemiology and Preventive Medicine, School of Public Health and Preventive Medicine, Monash University, Melbourne, Australia

[23] Department of Surgery and Perioperative Science, Umeå University, Umeå, Sweden

[24] Department of Neurosurgery, Medical School, University of Pécs, Hungary and Neurotrauma Research Group, János Szentágothai Research Centre, University of Pécs, Hungary

[25] Department of Medical Psychology, Universitätsklinikum Hamburg-Eppendorf, Hamburg, Germany

[26] Brain Physics Lab, Division of Neurosurgery, Dept of Clinical Neurosciences, University of Cambridge, Addenbrooke's Hospital, Cambridge, UK

[27] Neuro ICU, Fondazione IRCCS Cà Granda Ospedale Maggiore Policlinico, Milan, Italy

[28] ANZIC Research Centre, Monash University, Department of Epidemiology and Preventive Medicine, Melbourne, Victoria, Australia

[29] Department of Neurosurgery, Hospital of Cruces, Bilbao, Spain

[30] NeuroIntensive Care, Niguarda Hospital, Milan, Italy

[31] School of Medicine and Surgery, Università Milano Bicocca, Milano, Italy

[32] NeuroIntensive Care, ASST di Monza, Monza, Italy

[33] Department of Neurosurgery, Medical Faculty RWTH Aachen University, Aachen, Germany

[34] Department of Anesthesiology and Intensive Care Medicine, University Hospital Bonn, Bonn, Germany

[35] Department of Anesthesia & Neurointensive Care, Cambridge University Hospital NHS Foundation Trust, Cambridge, UK

[36] School of Public Health & PM, Monash University and The Alfred Hospital, Melbourne, Victoria, Australia

[37] Radiology/MRI department, MRC Cognition and Brain Sciences Unit, Cambridge, UK

[38] Institute of Medical Psychology and Medical Sociology, Universitätsmedizin Göttingen, Göttingen, Germany

[39] Oxford University Hospitals NHS Trust, Oxford, UK

[40] Intensive Care Unit, CHU Poitiers, Potiers, France

[41] University of Manchester NIHR Biomedical Research Centre, Critical Care Directorate, Salford Royal Hospital NHS Foundation Trust, Salford, UK

[42] Movement Science Group, Faculty of Health and Life Sciences, Oxford Brookes University, Oxford, UK

[43] Department of Neurosurgery, Antwerp University Hospital and University of Antwerp, Edegem, Belgium

[44] Department of Anesthesia & Intensive Care, Maggiore Della Carità Hospital, Novara, Italy

[45] Department of Neurosurgery, University Hospitals Leuven, Leuven, Belgium

[46] Department of Neurosurgery, Clinical centre of Vojvodina, Faculty of Medicine, University of Novi Sad, Novi Sad, Serbia

[47] Division of Anaesthesia, University of Cambridge, Addenbrooke's Hospital, Cambridge, UK

[48] Center for Stroke Research Berlin, Charité–Universitätsmedizin Berlin, corporate member of Freie Universität Berlin, Humboldt-Universität zu Berlin, and Berlin Institute of Health, Berlin, Germany

[49] Intensive Care Unit, CHR Citadelle, Liège, Belgium

[50] Department of Anaesthesiology and Intensive Therapy, University of Pécs, Pécs, Hungary

[51] Departments of Neurology, Clinical Neurophysiology and Neuroanesthesiology, Region Hovedstaden Rigshospitalet, Copenhagen, Denmark

[52] National Institute for Stroke and Applied Neurosciences, Faculty of Health and Environmental Studies, Auckland University of Technology, Auckland, New Zealand

[53] Department of Neurology, Erasmus MC, Rotterdam, the Netherlands

[54] Department of Anesthesiology and Intensive care, University Hospital Northern Norway, Tromso, Norway

[55] Department of Neurosurgery, Hadassah-hebrew University Medical center, Jerusalem, Israel

[56] Fundación Instituto Valenciano de Neurorrehabilitación (FIVAN), Valencia, Spain

[57] Department of Neurosurgery, Shanghai Renji hospital, Shanghai Jiaotong University/school of medicine, Shanghai, China

[58] Karolinska Institutet, INCF International Neuroinformatics Coordinating Facility, Stockholm, Sweden

[59] Emergency Department, CHU, Liège, Belgium

[60] Neurosurgery clinic, Pauls Stradins Clinical University Hospital, Riga, Latvia

[61] Department of Computing, Imperial College London, London, UK

[62] Department of Neurosurgery, Hospital Universitario 12 de Octubre, Madrid, Spain

[63] Department of Anesthesia, Critical Care and Pain Medicine, Medical University of Vienna, Austria

[64] Department of Public Health, Erasmus Medical Center-University Medical Center, Rotterdam, The Netherlands

[65] College of Health and Medicine, Australian National University, Canberra, Australia

[66] Department of Neurosurgery, Neurosciences Centre & JPN Apex trauma centre, All India Institute of Medical Sciences, New Delhi-110029, India

[67] Department of Neurosurgery, Erasmus MC, Rotterdam, the Netherlands

[68] Department of Neurosurgery, Oslo University Hospital, Oslo, Norway

[69] Division of Psychology, University of Stirling, Stirling, UK

[70] Division of Neurosurgery, Department of Clinical Neurosciences, Addenbrooke's Hospital & University of Cambridge, Cambridge, UK

[71] Department of Neurology, University of Groningen, University Medical Center Groningen, Groningen, Netherlands

[72] Neurointensive Care, Sheffield Teaching Hospitals NHS Foundation Trust, Sheffield, UK

[73] Salford Royal Hospital NHS Foundation Trust Acute Research Delivery Team, Salford, UK

[74] Department of Intensive Care and Department of Ethics and Philosophy of Medicine, Erasmus Medical Center, Rotterdam, The Netherlands

[75] Department of Clinical Neuroscience, Neurosurgery, Umeå University, Umeå, Sweden

[76] Hungarian Brain Research Program—Grant No. KTIA_13_NAP-A-II/8, University of Pécs, Pécs, Hungary

[77] Department of Anaesthesiology, University Hospital of Aachen, Aachen, Germany

[78] Cyclotron Research Center, University of Liège, Liège, Belgium

[79] Centre for Urgent and Emergency Care Research (CURE), Health Services Research Section, School of Health and Related Research (ScHARR), University of Sheffield, Sheffield, UK

[80] Emergency Department, Salford Royal Hospital, Salford UK

[81] Institute of Research in Operative Medicine (IFOM), Witten/Herdecke University, Cologne, Germany

[82] VP Global Project Management CNS, ICON, Paris, France

[83] Department of Anesthesiology-Intensive Care, Lille University Hospital, Lille, France

[84] Department of Neurosurgery, Rambam Medical Center, Haifa, Israel

[85] Department of Anesthesiology & Intensive Care, University Hospitals Southhampton NHS Trust, Southhampton, UK

[86] Cologne-Merheim Medical Center (CMMC), Department of Traumatology, Orthopedic Surgery and Sportmedicine, Witten/Herdecke University, Cologne, Germany

[87] Intensive Care Unit, Southmead Hospital, Bristol, Bristol, UK

[88] Department of Neurological Surgery, University of California, San Francisco, California, USA

[89] Department of Anesthesia & Intensive Care, M. Bufalini Hospital, Cesena, Italy

[90] Department of Neurosurgery, University Hospital Heidelberg, Heidelberg, Germany

[91] Department of Neurosurgery, The Walton centre NHS Foundation Trust, Liverpool, UK

[92] Department of Medical Genetics, University of Pécs, Pécs, Hungary

[93] Department of Neurosurgery, Emergency County Hospital Timisoara, Timisoara, Romania

[94] School of Medical Sciences, Örebro University, Örebro, Sweden

[95] Institute for Molecular Medicine Finland, University of Helsinki, Helsinki, Finland

[96] Analytic and Translational Genetics Unit, Department of Medicine; Psychiatric & Neurodevelopmental Genetics Unit, Department of Psychiatry; Department of Neurology, Massachusetts General Hospital, Boston, MA, USA

[97] Program in Medical and Population Genetics; The Stanley Center for Psychiatric Research, The Broad Institute of MIT and Harvard, Cambridge, MA, USA

[98] Department of Radiology, University of Antwerp, Edegem, Belgium

[99] Department of Anesthesiology & Intensive Care, University Hospital of Grenoble, Grenoble, France

[100] Department of Anesthesia & Intensive Care, Azienda Ospedaliera Università di Padova, Padova, Italy

[101] Dept. of Neurosurgery, Leiden University Medical Center, Leiden, The Netherlands and Dept. of Neurosurgery, Medical Center Haaglanden, The Hague, The Netherlands

[102] Department of Neurosurgery, Helsinki University Central Hospital

[103] Division of Clinical Neurosciences, Department of Neurosurgery and Turku Brain Injury Centre, Turku University Hospital and University of Turku, Turku, Finland

[104] Department of Anesthesiology and Critical Care, Pitié -Salpêtrière Teaching Hospital, Assistance Publique, Hôpitaux de Paris and University Pierre et Marie Curie, Paris, France

[105] Neurotraumatology and Neurosurgery Research Unit (UNINN), Vall d'Hebron Research Institute, Barcelona, Spain

[106] Department of Neurosurgery, Kaunas University of technology and Vilnius University, Vilnius, Lithuania

[107] Department of Neurosurgery, Rezekne Hospital, Latvia

[108] Department of Anaesthesia, Critical Care & Pain Medicine NHS Lothian & University of Edinburg, Edinburgh, UK

[109] Director, MRC Biostatistics Unit, Cambridge Institute of Public Health, Cambridge, UK

[110] Department of Physical Medicine and Rehabilitation, Oslo University Hospital/University of Oslo, Oslo, Norway

[111] Division of Orthopedics, Oslo University Hospital, Oslo, Norway

[112] Institue of Clinical Medicine, Faculty of Medicine, University of Oslo, Oslo, Norway

[113] Broad Institute, Cambridge MA Harvard Medical School, Boston MA, Massachusetts General Hospital, Boston MA, USA

[114] National Trauma Research Institute, The Alfred Hospital, Monash University, Melbourne, Victoria, Australia

[115] Department of Neurosurgery, Odense University Hospital, Odense, Denmark

[116] International Neurotrauma Research Organisation, Vienna, Austria

[117] Klinik für Neurochirurgie, Klinikum Ludwigsburg, Ludwigsburg, Germany

[118] Division of Biostatistics and Epidemiology, Department of Preventive Medicine, University of Debrecen, Debrecen, Hungary

[119] Department Health and Prevention, University Greifswald, Greifswald, Germany

[120] Department of Anaesthesiology and Intensive Care, AUVA Trauma Hospital, Salzburg, Austria

[121] Department of Neurology, Elisabeth-TweeSteden Ziekenhuis, Tilburg, the Netherlands

[122] Department of Neuroanesthesia and Neurointensive Care, Odense University Hospital, Odense, Denmark

[123] Department of Neuromedicine and Movement Science, Norwegian University of Science and Technology, NTNU, Trondheim, Norway

[124] Department of Physical Medicine and Rehabilitation, St. Olavs Hospital, Trondheim University Hospital, Trondheim, Norway

[125] Department of Neurosurgery, University of Pécs, Pécs, Hungary

[126] Division of Neuroscience Critical Care, John Hopkins University School of Medicine, Baltimore, USA

[127] Department of Neuropathology, Queen Elizabeth University Hospital and University of Glasgow, Glasgow, UK

[128] Dept. of Department of Biomedical Data Sciences, Leiden University Medical Center, Leiden, The Netherlands

[129] Department of Pathophysiology and Transplantation, Milan University, and Neuroscience ICU, Fondazione IRCCS Cà Granda Ospedale Maggiore Policlinico, Milano, Italy

[130] Department of Radiation Sciences, Biomedical Engineering, Umeå University, Umeå, Sweden

[131] Perioperative Services, Intensive Care Medicine and Pain Management, Turku University Hospital and University of Turku, Turku, Finland

[132] Department of Neurosurgery, Kaunas University of Health Sciences, Kaunas, Lithuania

[133] Intensive Care and Department of Pediatric Surgery, Erasmus Medical Center, Sophia Children's Hospital, Rotterdam, The Netherlands

[134] Department of Neurosurgery, Kings college London, London, UK

[135] Neurologie, Neurochirurgie und Psychiatrie, Charité–Universitätsmedizin Berlin, Berlin, Germany

[136] Department of Intensive Care Adults, Erasmus MC–University Medical Center Rotterdam, Rotterdam, the Netherlands

[137] icoMetrix NV, Leuven, Belgium

[138] Movement Science Group, Faculty of Health and Life Sciences, Oxford Brookes University, Oxford, UK

[139] Psychology Department, Antwerp University Hospital, Edegem, Belgium

[140] Director of Neurocritical Care, University of California, Los Angeles, USA

[141] Department of Neurosurgery, St. Olavs Hospital, Trondheim University Hospital, Trondheim, Norway

[142] Department of Emergency Medicine, University of Florida, Gainesville, Florida, USA

[143] Department of Neurosurgery, Charité–Universitätsmedizin Berlin, corporate member of Freie Universität Berlin, Humboldt-Universität zu Berlin, and Berlin Institute of Health, Berlin, Germany

[144] VTT Technical Research Centre, Tampere, Finland

[145] Section of Neurosurgery, Department of Surgery, Rady Faculty of Health Sciences, University of Manitoba, Winnipeg, MB, Canada

We are immensely grateful to our patients for helping us in our efforts to improve care and outcome for TBI. Furthermore, we would like to thank Monika Bullinger and Holger Muehlan for their ongoing and motivating support.

## Author Contributions

**Conceptualization:** Nicole von Steinbuechel, Marina Zeldovich.

**Data curation:** Marina Zeldovich.

**Formal analysis:** Johannes Vester, Marina Zeldovich.

**Funding acquisition:** Nicole von Steinbuechel, Suzanne Polinder.

**Methodology:** Nicole von Steinbuechel, Johannes Vester, Marina Zeldovich.

**Software:** Johannes Vester, Marina Zeldovich.

**Supervision:** Nicole von Steinbuechel.

**Visualization:** Marina Zeldovich.

**Writing – original draft:** Nicole von Steinbuechel, Katrin Rauen, Marina Zeldovich.

**Writing – review & editing:** Nicole von Steinbuechel, Katrin Rauen, Amra Covic, Ugne Krenz, Fabian Bockhop, Isabelle Mueller, Katrin Cunitz, Suzanne Polinder, Ewout W. Steyerberg, Johannes Vester, Marina Zeldovich.

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
