## [Decision Letter · Decision Letter 0]

2 Sep 2022

PONE-D-22-18258Sensitivity of outcome instruments in different patient groups after traumatic brain injury: results from the CENTER-TBI studyPLOS ONE

Dear Dr. Zeldovich,

Thank you for submitting your manuscript to PLOS ONE. After careful consideration, we feel that it has merit but does not fully meet PLOS ONE’s publication criteria as it currently stands. Therefore, we invite you to submit a revised version of the manuscript that addresses the points raised during the review process.

We look forward to receiving your revised manuscript.

Kind regards,

Jinglu Ai, M.D., Ph.D.

Academic Editor

PLOS ONE

Journal Requirements:

   "NO authors have competing interests"

5. One of the noted authors is a group or consortium [the CENTER-TBI participants and investigators]. In addition to naming the author group, please list the individual authors and affiliations within this group in the acknowledgments section of your manuscript. Please also indicate clearly a lead author for this group along with a contact email address.

Additional Editor Comments:

Your revised manuscript will be sent to the original reviewers for a second review. Please address adequately all the questions raised by the reviewers. Please pay special attention to the “sensitivity” question from reviewer 1.

Reviewers' comments:

Reviewer's Responses to Questions

**Comments to the Author**

1. Is the manuscript technically sound, and do the data support the conclusions?

Reviewer #1: No

Reviewer #2: Yes

2. Has the statistical analysis been performed appropriately and rigorously? 

Reviewer #1: No

Reviewer #2: Yes

3. Have the authors made all data underlying the findings in their manuscript fully available?

Reviewer #1: Yes

Reviewer #2: No

4. Is the manuscript presented in an intelligible fashion and written in standard English?

Reviewer #1: Yes

Reviewer #2: Yes

5. Review Comments to the Author

Reviewer #1: This manuscript addresses an important topic: given the complexity and multidimensionality of TBI outcomes, what are the most useful measures for capturing the most meaningful profile of this outcome scenario. Their primary metric for judging measure utility is sensitivity, defined as the ability of the measure to distinguish statistically among a set of a priori groups, defined by demographic, injury, and temporal recovery factors.

The manuscript reflects ambitious data collection and is rigorous in its methods and analyses. However, in my view, it suffers from a serious conceptual problem. Sensitivity is an important feature of any measure intended to distinguish among individuals at a point in time or within individuals across time or intervention. What a measure should be sensitive to is not standard. Yet in this study, each of the measures in question was assessed against the same predictor variables to judge its sensitivity.

The GOS-E was designed to capture the observed range of long-term functional outcome after TBI and to assess longitudinal progress toward that outcome. This it is not at all surprising that it is very sensitive to injury severity and injury recovery. In contrast, we know that depression is very common after mild TBI as well as severe and does not reliably remit over the course of recovery. We know that long-term outcome is sensitive to age. Does anxiety have a similar age-related pattern? If not, there’s no reason to expect it to be “sensitive to age.”

With these kinds of examples in mind, it does not seem meaningful to compare sensitivity to a standard set of predictors, across measures of different domains. One could, in principle, compare the sensitivity of the GOS-E to injury severity, on the one hand, to the sensitivity of the PHQ9 to pre/post antidepressant treatment, on the other. But how much functional recovery is equivalent to how much depression recovery? It seems that the only meaningful way to use the authors’ methodology, would be to compare the sensitivity or multiple measures within a domain to a set of predictors known to influence that domain.

Reviewer #2: This is a comprehensive manuscript evaluating the sensitivity of nine outcome instruments used in the large CENTER-TBI study at 3-, 6- and 12-months after TBI. The authors conclude that assessment of functional recovery status combined with generic HRQOL (PCS of the SF-12v2) and disease-specific HRQOL (QOLIBRI-OS), post-concussion symptoms (RPQ), and depression (PHQ-9) can provide a sensitive, comprehensive, yet time-efficient evaluation of the health status of individuals after TBI in different patient groups.

This is an important and timely paper describing data-driven recommendations for efficient use of outcome instruments at TBI follow-up timepoints. I have no major comments. The authors may consider including the following in the main text:

- Overall sample size and mild/moderate/severe TBI distribution in the Results section.

- In the Methods, clarify whether the authors utilize clinical cutoffs (rule-in/rule-out), ordinal analysis or scalar analysis of the scores on the outcome instruments.

- It does not appear that a sensitivity analysis was done for the GOSE/GOSE-Q. Perhaps I missed the rationale - is it because the GOSE is a gold standard measure for functional outcomes.

- Please comment briefly on the implications of using the GOSE vs. the GOSE-Q, and the QOLIBRI vs. the QOLIBRI-OS. Is there data to indicate that the proxy instrument (for GOSE-Q) or the more abbreviated form (QOLIBRI-OS) supply comparable / non-statistically significant differences to their more detailed questionnaire counterparts?

- Given the potential impact of this paper, do the authors think the current barriers (and imminent future directions) for post-TBI care lie in referring patients to proper follow-up after injury, or adoption of a core/recommended set of measures in the outcome setting? The authors can choose to include any part of this response in the discussion section.

6. PLOS authors have the option to publish the peer review history of their article (what does this mean?). If published, this will include your full peer review and any attached files.

Reviewer #1: **Yes: **John Whyte, MD, PhD

Reviewer #2: No

---

## [Author Response · Author response to Decision Letter 0]

6 Jan 2023

Reviewer #1: 

This manuscript addresses an important topic: given the complexity and multidimensionality of TBI outcomes, what are the most useful measures for capturing the most meaningful profile of this outcome scenario. Their primary metric for judging measure utility is sensitivity, defined as the ability of the measure to distinguish statistically among a set of a priori groups, defined by demographic, injury, and temporal recovery factors.

The manuscript reflects ambitious data collection and is rigorous in its methods and analyses. However, in my view, it suffers from a serious conceptual problem. Sensitivity is an important feature of any measure intended to distinguish among individuals at a point in time or within individuals across time or intervention. What a measure should be sensitive to is not standard. Yet in this study, each of the measures in question was assessed against the same predictor variables to judge its sensitivity.

Response: Dear Reviewer, thank you for your valuable feedback. We are going to explain why we have chosen this approach and why we think that our study results can provide an added value for both clinical practice and TBI research. 

In the present study, we aimed to assess the multidimensional sensitivity of outcomes instruments after TBI, which is indeed not yet a standard approach. To date, only the sensitivity of individual instruments used in the field of TBI has been assessed, if at all. Given the interdependencies of the different outcomes reflected in the intercorrelations (see Figure S1), we believe that the use of a multivariate approach is appropriate and necessary in order to assess sensitivity. 

We have tried to make it clear in the Introduction section (rationale of the study):

Page 5, lines 120 ff.:

“Given the impact of TBI on different domains of health and life, and considering the heterogeneity of potential risk and protective factors, a sensitive multidimensional approach is needed to identify the short- and long-term effects of the injury. To date, only the sensitivity of individual instruments used in the field of TBI has been assessed, if at all. Systematic analyses of the multivariate sensitivity of the instruments measuring outcome domains concerning patient groups selected a priori in the field of TBI is still scarce. 

To fill this gap, the sensitivity of the PROMs that assess these domains needs to be investigated with reference to several relevant patient groups, which are known from the TBI literature, and with reference to functional recovery. We therefore aim to investigate the multidimensional cross-sectional sensitivity of selected outcome instruments using a patient-centered, group-based diagnostic approach. This approach includes the analysis of sensitivity at three different time points (i.e., 3, 6, and 12 months) as the sensitivity of the instruments can differ depending on the time of assessment post TBI.” 

Furthermore, sensitivity has not yet been analyzed for an ensemble of multidimensional outcomes considering predictors or risk factors (i.e., patient groups) that are known from the literature. We believe that the use of such predictors is an accepted way to examine sensitivity from a psychometric perspective and that the results of these analyses can help clinicians and researchers select the most sensitive instrument for their specific question. 

We have added the following information to the Introduction section:

Page 5, lines 109 ff.:

“Overall, lower functional recovery, reduced generic and TBI-specific HRQOL, and higher symptom burden (i.e., anxiety, depression, post-traumatic stress disorder, and post-concussion symptoms) were repeatedly associated with female gender (21) (29), higher age (44) (45) (46), lower education (25) (47) (48), the presence of premorbid psychological problems (4) (45) (49) (50) (51), being discharged home from the emergency room (52) or being admitted to the ICU (43) (53) (54), as well as having more severe extracranial injuries or polytrauma (38) (43) (53) (55) (56) (57), and higher TBI severity (24) (38) (56) (58) (59) (see Table in S1 Table for a more detailed overview). Hence, analyzing the sensitivity of outcome instruments to patient groups based on these characteristics can assist in selecting the appropriate instruments. This may contribute to better clinical decision-making and personalized treatment.”

Below you will find our detailed responses to your suggestions and questions regarding the suitability of the chosen approach.

The GOS-E was designed to capture the observed range of long-term functional outcome after TBI and to assess longitudinal progress toward that outcome. This it is not at all surprising that it is very sensitive to injury severity and injury recovery. 

Response: Firstly, although the GOSE is seen as the core measure for outcome assessment after TBI according to the Common Data Elements (CDE) recommendations (https://www.commondataelements.ninds.nih.gov/Traumatic%20Brain%20Injury, last accessed on 23.11.22), there are studies which have found it to suffer from item redundancy and inefficiency, and to produce ceiling effects (please, see Introduction, p. 4, lines 75 ff.). We therefore believe that additional evidence of the GOSE’s sensitivity as a core outcome measure can further strengthen its role in clinical practice and research. Secondly, we have used the GOSE as a reference, to consolidate the clinical relevance of our analyses. 

We have added the following to the Methods section:

Page 13, lines 308 ff.:

“This approach was chosen to strengthen the evidence for the GOSE as a core measure in the field of TBI, to review the criticisms formulated regarding its applicability (15) (16) (17) (18), and to consolidate the clinical relevance of the analyses in the present study.”

In addition, the GOSE was not designed to measure outcomes other than functional recovery. Considering that multiple outcome domains are often affected after a TBI (see Introduction, p. 4, line 86 ff.) and that these domains are correlated (see Figure S1), it is important to examine them simultaneously and subsequently include them in the diagnostic process. 

We stated the following in the Methods and Results:

Methods; Page 12, lines 280 ff.:

“Spearman correlations investigated the strength of associations between the outcome domains. Effect sizes were classified as being small (.10), medium (.30), and large (.50) (85) (86). Medium to high associations between the outcome instruments warrant conducting multivariate analyses.”

Results; Page 16, lines 387 ff.:

“The outcome domains were moderately to highly correlated, except for the MCS and PCS of the SF-36v2/-12v2, which had a low correlation (< .30) with each other, justifying and requiring the use of multidimensional analyses. For details, see Fig in S1 Fig.”

Furthermore, we have provided the following additional information in the Discussion section:

Discussion; Page 22, lines 531 ff.:

“Recent studies (94,96,97) suggest that the use of a single outcome instrument, such as the GOSE, may not provide a comprehensive picture of patients’ health status. For a comprehensive representation of a patient’s clinical picture, it is recommended that multidimensional outcome assessments are performed. The medium to high sensitivity of the PROMs to the recovery status of the patients supports the approach of complementing the sole GOSE assessment by other clinically relevant outcome measures. Based on the results of our analyses, the use of the physical generic HRQOL component of the SF-36v2/-12v2 instruments (PCS), the disease-specific HRQOL measures (QOLIBRI/-OS) and the RPQ to assess post-concussion symptoms can be recommended to provide a sensitive and valid multidimensional assessment of outcomes and impairments in combination with the GOSE.”

In contrast, we know that depression is very common after mild TBI as well as severe and does not reliably remit over the course of recovery. We know that long-term outcome is sensitive to age. Does anxiety have a similar age-related pattern? If not, there’s no reason to expect it to be “sensitive to age.” With these kinds of examples in mind, it does not seem meaningful to compare sensitivity to a standard set of predictors, across measures of different domains.

Response: Thank you for raising this important issue. Yes, the groups selected a priori are relevant for all outcomes considered in our study. To provide a more detailed overview of their importance, we have presented additional results of a literature search in Table S1. To obtain robust results, systematic reviews and meta-analyses were preferred over other study types. For studies with cross-sectional and longitudinal designs, non-CENTER-TBI studies were preferred over CENTER-TBI studies to provide empirical evidence based on data from other projects. The results underline that all outcomes investigated follow the same pattern with respect to the groups selected. Therefore, it seems reasonable to examine all outcomes in relation to all selected a priori patient groups. We have provided a comprehensive overview of the studies in S1 Table and included the following information in the Introduction and Methods section:

Introduction; Page 4, lines 93 ff.:

“A wide range of literature, including systematic reviews, meta-analyses, cross-sectional and longitudinal multinational studies, has addressed the question of identifying protective and risk factors for TBI outcome. Previous research has shown that outcomes may be influenced by the sociodemographic and clinical characteristics of individuals affected by TBI. There are controversial results concerning men or women having better outcomes, possibly in association with the premorbid health status or injury severity (28) (29). In addition, individuals aged 65 years and older are at a higher risk of mortality and unfavorable outcomes after TBI compared to younger individuals (30). Individuals with a lower pre-injury level of education tend to have worse cognitive outcomes after TBI and lower probability of a satisfactory return to work and life (31). Furthermore, the premorbid health status (32) (33) (34) and injury-related factors (e.g., different mechanisms of brain trauma (35) (36) (37), severity of brain injury (38) (39), or presence of extracranial injuries or major trauma (40)) may influence the outcome after TBI. A comparison of outcomes of uncomplicated and complicated mild TBI patients based on the combination of the Glasgow Coma Scale (GCS) (41) and findings from computed tomography (CT) scans (42) has shown that individuals after a complicated mild TBI had worse functional outcomes, decreased HRQOL, and a higher symptom burden compared with those who had experienced an uncomplicated mild TBI (39) (43). Overall, lower functional recovery, reduced generic and TBI-specific HRQOL, and higher symptom burden (i.e., anxiety, depression, post-traumatic stress disorder, and post-concussion symptoms) were repeatedly associated with female gender (21) (29), higher age (44) (45) (46), lower education (25) (47) (48), the presence of premorbid psychological problems (4) (45) (49) (50) (51), being discharged home from the emergency room (52) or being admitted to the ICU (43) (53) (54), as well as having more severe extracranial injuries or polytrauma (38) (43) (53) (55) (56) (57), and higher TBI severity (24) (38) (56) (58) (59) (see Table in S1 Table for a more detailed overview). Hence, analyzing the sensitivity of outcome instruments to patient groups based on these characteristics can assist in selecting the appropriate instruments.”

Methods; Page 8, lines 183 ff.:

“The sensitivity of the outcome instruments was examined using a priori selected groups covering sociodemographic, premorbid, and injury-related characteristics derived from previous studies. Table in S1 Table provides an overview of these characteristics influencing outcome domains (i.e., functional recovery, generic and disease-specific HRQOL, anxiety, depression, PTSD, and post-concussion symptoms) after TBI. The selected factors were found to be both significant and clinically relevant in several studies concerning a single outcome domain after TBI. For this reason, considering them when selecting instruments may have substantial benefits concerning diagnosis and treatment planning. Our multivariate analyses were therefore stratified according to the following sociodemographic characteristics: sex (male/female), age (<65/≥ 65 years), and education (primary and less/at least secondary). Premorbid health status and injury-related characteristics were assessed using the following information collected at the time of study enrollment: individuals’ psychological health status before the injury (emotional disorders, treatment for any mental health problems, or hospital admission for psychiatric reasons; absent/present), clinical pathways (ER/ward/ICU), and total injury severity score (ISS; with the cut-off values <10 indicating mild injury vs. ≥10 including moderate, severe and profound injuries) (70) as measured by the Abbreviated Injury Scale (AIS) (71). TBI severity was determined based on the GCS together with the information on CT findings, resulting in the following groups: uncomplicated mild (GCS ≥ 13 and no CT abnormalities), complicated mild (GCS ≥ 13 and visible CT abnormalities), moderate (9 ≤ GCS ≤ 12), and severe (GCS ≤ 8) TBI.”

Furthermore, to illustrate our results, we have included the example of a forest plot concerning the sensitivity of the PROMs in differentiating between good recovery (GOSE 7-8) and moderate disability (GOSE 5-6) for complicated mild TBI three months post-injury (Fig 3). Forest plots created for all a priori group analyses are provided in the supplemental material (S2 Text).

Results; P. 19, lines 444 ff.:

“Fig 3 provides an example concerning which PROM can distinguish between good recovery (GOSE 7-8) and moderate disability (GOSE 5-6) in complicated mild TBI, and to what extent. Overall, the entire ensemble of PROMs displayed a high sensitivity in differentiating between the patient groups selected a priori. The pooled effect was slightly above 0.29, CI95% [0.25, 0.33]. The group with good recovery displayed better outcomes compared with the group with the less favorable recovery, which was reflected by the high MW effect sizes. The RPQ, the PCS of the SF-36v2, and the QOLIBRI-OS presented the strongest effects and were thus most sensitive to the differences concerning the recovery status after complicated mild TBI. All other instruments had medium sensitivity, with CIs not exceeding the cut-off of 0.36. This indicated that the effect was stable medium with a 95% probability. An exception was the MCS of both forms of the SF instruments, where the lower CI cut-off was in the low sensitivity range (i.e., below 0.36). For details concerning the pooled (i.e., combined) MW effect sizes of the PROMs and respective effects in different groups of interest, see Text in S2 Text.”

One could, in principle, compare the sensitivity of the GOS-E to injury severity, on the one hand, to the sensitivity of the PHQ9 to pre/post antidepressant treatment, on the other.

Response: Thank you for this suggestion. In our opinion, this approach would not allow outcomes to be considered simultaneously and thus would not be appropriate for assessing the multidimensional sensitivity of the outcome instruments. As shown in Table S1, different levels of injury severity contribute significantly to all selected outcomes domains. Therefore, we believe that this kind of comparison is appropriate. Because our data set did not contain information on medical or pharmacological treatments, we were not able to include this in the analyses. We have noted in the study limitations the missing information that could be incorporated to further investigate the sensitivity of the outcomes after TBI. 

Discussion (Limitation subsection); P. 27, lines 647 ff.:

“In our study, the information regarding psychiatric problems before and after the TBI of the participants was based solely on self-reported data. Standardized clinical diagnoses of depression, anxiety, and PTSD as well as information on psychopharmacological treatment effects would contribute to a more precise differentiation, providing valuable directions for future studies. The sensitivity to detecting drug effects can however only be evaluated once the sensitivity of instruments to relevant predictors or risk factors has already been established.” 

But how much functional recovery is equivalent to how much depression recovery? 

Response: We agree that the extent to which the degree of functional recovery corresponds to the degree of depression recovery is of great importance. However, this is beyond the scope of the present study. In our ongoing investigation, we are focusing precisely on this topic by creating longitudinal outcome trajectories and linking them to the degree of functional recovery. We have added the following point to the discussion:

P. 28, lines 662 ff.:

“To gain better insight into the course of recovery after a TBI, future research should examine how multiple psychological and symptom-related PROMs are associated with trajectories of the functional recovery status.”

It seems that the only meaningful way to use the authors’ methodology, would be to compare the sensitivity of multiple measures within a domain to a set of predictors known to influence that domain. 

Response: Your suggestion to test multiple measures measuring the same outcome domain (e.g., GAD-7, HADS, and BDI-II to assess depressive symptoms) is interesting in order to determine which questionnaire has the highest sensitivity in this given domain. However, this is beyond the scope of the present study. Our goal was to address the gap in multidimensional sensitivity of instruments assessing the most commonly affected outcomes after a TBI which was not provided yet. We have added the following to the Introduction section:

P. 5, lines 122 ff.:

“To date, only the sensitivity of individual instruments used in the field of TBI has been assessed, if at all. Systematic analyses of the multivariate sensitivity of the instruments measuring outcome domains concerning patient groups selected a priori in the field of TBI is still scarce. 

To fill this gap, the sensitivity of the PROMs that assess these domains needs to be investigated with reference to several relevant patient groups, which are known from the TBI literature, and with reference to functional recovery.”

To investigate the sensitivity of multiple instruments measuring one outcome domain, we would recommend using another approach instead of Wei-Lachin analyses (e.g., methods comparable to meta-analytic techniques (1,2)). 

1. Gilbody S, Richards D, Brealey S, Hewitt C. Screening for Depression in Medical Settings with the Patient Health Questionnaire (PHQ): A Diagnostic Meta-Analysis. J GEN INTERN MED. 2007 Oct 12;22(11):1596–602. 

2. Zeldovich M, Alexandrowicz RW. Comparing outcomes: The Clinical Outcome in Routine Evaluation from an international point of view. Int J Methods Psychiatr Res [Internet]. 2019 Sep [cited 2020 Aug 13];28(3). Available from: https://onlinelibrary.wiley.com/doi/abs/10.1002/mpr.1774

We hope that we have succeeded in addressing your questions and incorporating your suggestions into our text so that we can better justify the selection of our study design and the methods used. By incorporating your feedback, we hope to present the relevance and validity of our study even more convincingly. 

 

Reviewer #2: 

This is a comprehensive manuscript evaluating the sensitivity of nine outcome instruments used in the large CENTER-TBI study at 3-, 6- and 12-months after TBI. The authors conclude that assessment of functional recovery status combined with generic HRQOL (PCS of the SF-12v2) and disease-specific HRQOL (QOLIBRI-OS), post-concussion symptoms (RPQ), and depression (PHQ-9) can provide a sensitive, comprehensive, yet time-efficient evaluation of the health status of individuals after TBI in different patient groups.

This is an important and timely paper describing data-driven recommendations for efficient use of outcome instruments at TBI follow-up timepoints. I have no major comments. The authors may consider including the following in the main text:

Response: Dear Reviewer, thank you very much for your valuable, very encouraging feedback. We have added the required information. Please, find our detailed responses below. 

- Overall sample size and mild/moderate/severe TBI distribution in the Results section.

Response: Thank you for mentioning this point. Since the number of participants included in our study varied dependent on outcome, we cannot state the total sample size. The sample attrition plot (Figure 2) provides an overview of response rates per instrument and time point. We have added the following summary in the Results section:

P. 15, line 364 ff.:

“Depending on the outcome instrument and the time of the assessment, the sample size for the outcome assessments varied from N = 2088 (GAD-7) to N = 2842 (GOSE/Q) at 3 months, from N = 2181 (GAD-7) to N = 2760 (GOSE/-Q) at six months, and N = 1437 (SF-36v2) to N = 1977 (GOSE/-Q) at twelve months. Participants were predominately male (> 60%), younger than 65 years of age (approx. 75%) and had at least a secondary school certificate (approx. 70%). The majority reported having no premorbid psychological problems (> 50%). They had mainly suffered an uncomplicated (around 30%) or a complicated mild TBI (around 30%), followed by severe (10% to 19%) and moderate (5% to 8%) TBI. Patients were mostly admitted to an ICU (> 40%) and had an ISS > 10 (> 60%). Sample characteristics for each instrument and time point are shown in Table in S2 Table. Fig 2 provides information on the sample sizes.”

- In the Methods, clarify whether the authors utilize clinical cutoffs (rule-in/rule-out), ordinal analysis or scalar analysis of the scores on the outcome instruments.

Response: For the PROMs, the cut-offs we used were suggested in previous studies or questionnaire manuals to determine impaired outcomes. Functional outcome was considered impaired if recovery was judged to be incomplete (i.e., a GOSE/-Q score <7). To account for the nature of GOSE scores, all statistical approaches chosen were appropriate for ordinal data (e.g., Wei-Lachin analyses or Spearman correlation analyses). 

We have added the required information in the Methods section:

P. 12, line 279 ff.:

“To account for the nature of the GOSE ratings, all statistical approaches chosen were appropriate for ordinal data.”

P. 13, line 300 ff.:

“Analyses were conducted using the total scores of the outcome instruments, except for the SF-36v2/-12v2, in which PCS and MCS were considered separately.”

P. 14, line 331 ff.:

“For the PROMs, impaired outcomes were determined using clinical cutoffs reported in previous studies (see description of instruments). For the GOSE/-Q, an outcome was considered impaired if recovery was rated as not complete (i.e., a GOSE/-Q score < 7).”

- It does not appear that a sensitivity analysis was done for the GOSE/GOSE-Q. Perhaps I missed the rationale - is it because the GOSE is a gold standard measure for functional outcomes.

Response: Thank you for this comment. The sensitivity analyses were also performed for the GOSE. First, we included the GOSE/-Q in the analyses to examine the sensitivity of all outcome instruments administered in the present study (please, see the description in the Method section: p. 13, line 303 ff). Second, we used it as a clinical reference (please, see p. 13, line 307 ff). In the Results section (“Sensitivity of all outcome instruments”, p. 16, line 391 ff.), we have briefly summarized the results. Tables in the supplemental material (Tables S4-S5) contain all further details. 

P. 16, line 391 ff.:

“The GOSE/-Q displayed the highest sensitivity across all patient groups and time points. The PCS and MCS of the SF-36v2/-12v2, the QOLIBRI/-OS, and the RPQ were most sensitive in the group comparisons at one or more point in time (see Table in S3 Table). For more details on the MW effect sizes, see Tables in S4-S5 Tables.”

- Please comment briefly on the implications of using the GOSE vs. the GOSE-Q, and the QOLIBRI vs. the QOLIBRI-OS. Is there data to indicate that the proxy instrument (for GOSE-Q) or the more abbreviated form (QOLIBRI-OS) supply comparable / non-statistically significant differences to their more detailed questionnaire counterparts?

Response: Thank you for your comment on this topic. We had already briefly commented in the discussion section that the short versions of the HRQOL instruments function in a similar way to the longer ones. We have additionally added a note on exactly which instruments we mean (i.e., QOLIBRI/-OS to measure TBI-specific and SF-36v2/.12v2 to measure overall HRQOL):

Comment on QOLIBRI/-OS; P. 23, line 541 ff.:

“The short forms of the HRQOL instruments (i.e., QOLIBRI/-OS measuring TBI-specific and SF-36v2/-12v2 measuring generic HRQOL) showed comparable sensitivity to their longer versions. Therefore, they could be useful in both routine clinical assessment and research to reduce patient burden and to save assessment time.”

Unfortunately, we cannot provide any comment on the administration of the GOSE vs. GOSE-Q since we had to use combined information with missing GOSE values substituted by the information derived either from the GOSE-Q or the interviewer ratings provided centrally. However, there is empirical evidence for a good agreement between the GOSE and its questionnaire version (Horton et al., 2021). Since we believe that this point is important, we have added this information to the study limitations:

Comment on GOSE/-Q; P. 27, line 653 ff.:

“Furthermore, the functional recovery status of the study participants was determined using the GOSE, with missing values substituted based on the GOSE-Q and/or clinical assessments. Therefore, it is not possible to compare the sensitivity of the GOSE interview with its questionnaire version. In a recent study, GOSE ratings showed good agreement with GOSE-Q scores and a similar association with other outcomes after TBI (111). However, future studies should further address the sensitivity of the two GOSE forms to provide more evidence for their applicability and mutual substitution.”

 - Given the potential impact of this paper, do the authors think the current barriers (and imminent future directions) for post-TBI care lie in referring patients to proper follow-up after injury, or adoption of a core/recommended set of measures in the outcome setting? The authors can choose to include any part of this response in the discussion section. 

Response: Thank you very much for this important advice. We believe that both points mentioned are of utmost importance for post-TBI care. For proper follow-up (including the provision of appropriate rehabilitation services), the basis for valid, reliable, and sensitive measurement instruments capable of detecting potential impairment should first be demonstrated. Therefore, we strongly recommend complementing the information obtained from the GOSE with results from the most sensitive PROMs. Upon your suggestion, we have supplemented our recommendations concerning the multidimensional assessment and selection of instruments by the following information in the Discussion part:

Discussion; P. 26, line 614 ff.:

“To further develop post-TBI care, treatment, and rehabilitation, an assessment of potential deficits should be conducted as early as possible and longitudinally using reliable, valid, and sensitive instruments that measure the consequences of the TBI multidimensionally in all relevant health states and life domains. The PROMs analyzed in the present study provide this basis. The use of these instruments in combination with the GOSE would again allow timely diagnosis and treatment at follow-up visits, which should be performed at several time points at least up to one year after TBI to help to control, prevent, or reduce the manifestation of symptoms in various outcome domains.”

Recommendations; P. 28, line 668 ff.:

“For a sensitive, reliable, economic, yet comprehensive assessment of outcomes after TBI, the evaluation of the recovery status should be combined with self-reports on physical generic HRQOL (e.g., PCS of the SF-12v2), disease-specific HRQOL (e.g., QOLIBRI-OS), and post-concussion symptoms (RPQ). If time and patient burden allow, the severity of major depression should additionally be assessed with the PHQ-9 if it was not diagnosed clinically. The suggested, relatively short multidimensional yet comprehensive outcome assessment of individuals after TBI of all severities may help to evaluate treatment effects sensitively and tailor interventions and care after TBI.”

---

## [Editor Report · Decision Letter 1]

10 Jan 2023

Sensitivity of outcome instruments in a priori selected patient groups after traumatic brain injury: results from the CENTER-TBI study

PONE-D-22-18258R1

Dear Dr. Zeldovich,

We’re pleased to inform you that your manuscript has been judged scientifically suitable for publication and will be formally accepted for publication once it meets all outstanding technical requirements.

Kind regards,

Jinglu Ai, M.D., Ph.D.

Academic Editor

PLOS ONE
---

## [Editor Report · Acceptance letter]

27 Feb 2023

PONE-D-22-18258R1 

Sensitivity of outcome instruments in a priori selected patient groups after traumatic brain injury: results from the CENTER-TBI study 

Dear Dr. Zeldovich:

I'm pleased to inform you that your manuscript has been deemed suitable for publication in PLOS ONE. Congratulations! Your manuscript is now with our production department. 

Kind regards, 

on behalf of

Dr. Jinglu Ai 

Academic Editor

PLOS ONE